# Regulatory Elements for Gene Therapy of Epilepsy

**DOI:** 10.3390/cells14030236

**Published:** 2025-02-06

**Authors:** Ekaterina Chesnokova, Natalia Bal, Ghofran Alhalabi, Pavel Balaban

**Affiliations:** 1Laboratory of Cellular Neurobiology of Learning, Institute of Higher Nervous Activity and Neurophysiology of the Russian Academy of Sciences, Moscow 117485, Russia; chesnokova@ihna.ru (E.C.); pmbalaban@gmail.com (P.B.); 2Laboratory of Molecular Neurobiology, Institute of Higher Nervous Activity and Neurophysiology of the Russian Academy of Sciences, Moscow 117485, Russia; ghofranalhalabi123@gmail.com

**Keywords:** epilepsy, gene therapy, viral expression vectors, cis-regulatory elements, promoters, enhancers, glutamatergic neurons, GABAergic neurons, astrocytes, promoter silencing

## Abstract

The problem of drug resistance in epilepsy means that in many cases, a surgical treatment may be advised. But this is only possible if there is an epileptic focus, and resective brain surgery may have adverse side effects. One of the promising alternatives is gene therapy, which allows the targeted expression of therapeutic genes in different brain regions, and even in specific cell types. In this review, we provide detailed explanations of some key terms related to genetic engineering, and describe various regulatory elements that have already been used in the development of different approaches to treating epilepsy using viral vectors. We compare a few universal promoters for their strength and duration of transgene expression, and in our description of cell-specific promoters, we focus on elements driving expression in glutamatergic neurons, GABAergic neurons and astrocytes. We also explore enhancers and some other cis-regulatory elements currently used in viral vectors for gene therapy, and consider future perspectives of state-of-the-art technologies for designing new, stronger and more specific regulatory elements. Gene therapy has multiple advantages and should become more common in the future, but there is still a lot to study and invent in this field.

## 1. Introduction

### 1.1. The Problem of Drug Resistance in Epilepsy: Gene Therapy as a Novel Alternative to Antiseizure Drugs

A seizure is defined as “a transient occurrence of signs and/or symptoms due to abnormal excessive or synchronous neuronal activity in the brain”. The symptoms in question are diverse. Some common examples are impaired awareness, subjective sensations (hallucinations, sudden emotions) and motor manifestations (automatisms, increased muscle tonus, spasms). Seizure onset may be focal (originating within networks limited to one hemisphere) or generalized (originating at some point within, and rapidly engaging, bilaterally distributed networks) [1]. Onset is determined based on the clinical features, electroencephalography (EEG) and neuroimaging findings. Seizures may be caused by diverse conditions affecting normal brain function [2]. However, a single seizure provoked by an obvious damaging factor (like concussion or fever) would not lead to a diagnosis of epilepsy. According to the current documents of the International League Against Epilepsy (ILAE), epilepsy is defined by any of the following conditions: (1) at least two unprovoked (or reflex) seizures occurring >24 h apart; (2) one unprovoked seizure and a probability of further seizures similar to the general recurrence risk (at least 60%) after two unprovoked seizures, occurring over the next 10 years; (3) diagnosis of an epilepsy syndrome [3]. An epilepsy syndrome is defined as “a characteristic cluster of clinical and EEG features, often supported by specific etiological findings (structural, genetic, metabolic, immune, and infectious)”. These syndromes are classified based on age at onset and on syndrome type (generalized, focal, focal and generalized, and syndromes associated with developmental and/or epileptic encephalopathy (DEE) or progressive neurological deterioration) [4].

Epilepsy affects an estimated 50 million people worldwide and is one of the most prevalent neurological conditions. People with epilepsy have increased mortality (attributable to both seizure-related and unrelated causes), reduced quality of life, and lower employment and productivity [5]. Epilepsy is associated with increased odds of anxiety disorders, suicidal thoughts and major depression [6]. Other comorbidities include dementia, migraine, heart disease, peptic ulcers, and arthritis [7].

Approximately 30–40% of people with epilepsy are drug-resistant. For these patients, the best hope for an effective treatment is resective surgery. Unfortunately, this is suitable for fewer than 5% because of possible adverse effects [8]. Also, it is often difficult to localize or resect the epileptogenic locus completely [9]. Novel treatments of drug-resistant epilepsy are being developed, and gene therapy is among the most promising approaches. Gene therapy could represent an alternative to resective surgery; the use of gene therapy allows to modify the behavior of neurons in the epileptic focus without disrupting the physiological function of a brain region. This is particularly important for foci that are not normally resected because they are necessary for crucial brain functions such as movement, vision and speech. Alternatively, gene therapy may be used as an additional treatment to improve the success rate of surgery [8]. To date, at least four studies of gene therapy in epilepsy have reached the clinical stage, according to the clinicaltrials.gov database [10,11,12,13]. These studies test different ways to decrease pathological neuronal network activity (summarized in Table 1).

Varying approaches to treating epilepsy by gene therapy may be based on different regulatory genetic elements that control the expressions of different proteins in either specific cell types or multiple cell types, in a specific state or at specific time intervals.

Even though epilepsy is not always inherited, more than 500 genes associated with it have been identified [2]. Some kinds of epilepsy are so-called monogenic epilepsies, each caused by a mutation in a single gene. Such genes usually encode ion channels, synaptic receptors or transporter proteins. However, some forms of epilepsy have a complex pattern of inheritance involving mutations in multiple genes and their synergistic interactions with environmental factors [9]. Experiments with bulk RNA-Seq analysis of resected epileptic tissue showed relatively minor transcriptomic changes, but recent advances in high-resolution methods have given us a better perspective on specific cell types and signaling pathways involved in the pathogenesis of epilepsy. Pfisterer et al. performed single-nucleus transcriptomics analyses of >110,000 neuronal transcriptomes derived from temporal cortex samples of temporal lobe epilepsy (TLE) and non-epileptic subjects. The most notable gene expression changes associated with epilepsy were found in a few groups of principal neurons (L5-6_Fezf2 and L2-3_Cux2) and GABAergic interneurons (somatostatin (SST)+ and parvalbumin (PV)+ types). Importantly, the most affected cell types can be grouped based on the commonality of transcriptional changes, indicating that they belong to the same circuit that might underlie epileptogenesis. One of the most dysregulated pathways in epilepsy is glutamate signaling. Genes coding for α-amino-3-hydroxy-5-methyl-4-isoxazolepropionic acid (AMPA) glutamate receptor auxiliary subunits might be the strongest contributors to epileptogenesis, being upregulated layer-wise [16]. The genomic and transcriptomic data described above are useful for choosing new targets for gene therapy, especially considering that cell type-specific transgene expression is now possible, and instruments for more precise cell type targeting are being actively developed.

The most obvious idea for gene therapy is to substitute alleles with loss-of-function mutations when the mutation is known. For example, Dravet syndrome (DS) is a type of monogenic epilepsy, a DEE associated with monoallelic loss-of-function mutations in the *SCN1A* gene. This gene encodes the Na_V_1.1 subunit of the primary voltage-gated sodium channel responsible for the generation of action potentials in GABAergic interneurons [14,17]. This is an example of a promising target for gene therapy, wherein the normal allele of this gene is expressed exogenously from a viral vector. Indeed, there are two ongoing clinical trials testing the benefits of exogenously expressed *SCN1A* [10,11]. It should be noted that, in theory, similar results may be achieved with another voltage-gated sodium channel (human, from other species, or fully artificial) expressed using a viral vector.

However, for most patients, identifying a single pathogenic mutation in the etiology of epilepsy is challenging, and it may be more rational to target symptoms to reduce clinical manifestations [18]. In this case, the choice of a therapeutic transgene is intended to compensate for the neurophysiological imbalance in epilepsy. The ectopic expression of ion channels may be used to decrease seizures [19]. For example, a clinical study of an epilepsy gene therapy using a lentiviral vector expressing an engineered potassium channel is currently at its first stage [12]. Genes encoding inhibitory peptides like galanin, neuropeptide Y (NPY), SST or dynorphin are also suggested for this purpose [9,20,21,22,23]. Using secreted factors like these peptides has an advantage, in that only the cells that express its receptors would be affected by it, and it does not matter what cell types produce it. Thus, the ability of the vector to target specific cell types becomes less important [24]. There are ways of increasing secretion efficiency; for example, the peptide of interest may be fused with the fibronectin secretory signal sequence (FIB) [21,25,26]. Genes encoding glial proteins, for example, adenosine kinase [27], are also considered to be potential targets for epilepsy gene therapy. Another approach involves manipulating the expression levels of endogenous genes rather than introducing new transcripts. For this, a CRISPR-based method that activates a genomic promoter (CRISPRa) may be used. The CRISPRa system is versatile because the targeted gene can be switched simply by changing the short guide RNA (sgRNA), which does not significantly affect the vector properties. It is also theoretically possible to combine multiple sgRNAs in the same vector, targeting multiple genes. This approach is also free from problems with ensuring the normal splicing and post-transcriptional processing of exogenous therapeutic mRNAs [28], even though Cas9-like proteins themselves are exogenous, and in some circumstances may be toxic [29] or immunogenic.

There are a few different types of viral vectors used in gene therapy, with different genome sizes, abilities to cross the blood–brain barrier (BBB), tropisms for different cell types, and undesired side effects such as immunogenicity or the risk of insertional mutagenesis. In studies of neurological disorders, the most commonly used vectors are recombinant adeno-associated viruses (AAV), lentiviruses (LV) or herpes simplex viruses (HSV). Their relative advantages are described in detail in [8,24]. The majority of gene therapy programs targeted at the central nervous system (CNS) use AAV vectors; a detailed comparison of different AAV serotypes, their biodistribution after infection and their tropism (tissue and cell-type selectivity) may be found in [30,31,32]. It must be noted that HSVs are more often used as research tools for animal experiments than as gene therapy vectors [9]. Adenoviral vectors were used in some initial studies of viral transgene delivery [33,34,35], but were then mostly avoided due to their high immunogenicity. However, recently, modified and improved adenoviruses have been proposed as an alternative viral platform for transgene delivery in the CNS. Namely, high-capacity adenoviral vectors (HC-AdVs) [36] and canine adenovirus type 2 (CAV-2) [37] were used in studies with DS models. In contrast with early versions, HC-AdVs are devoid of all viral coding genes [36]. There are also nonviral vectors, which are lipid-based or polymer-based nanoparticles, but these are still far from clinical practice [9]. We should also mention a novel technology called “encapsulated cell biodelivery” that is suitable for secreted factors. Genetically modified cells that secret a therapeutic agent are encapsulated in a polymer membrane and then injected into the brain. The pores of the membrane allow the secreted agent to diffuse, but are small enough to protect the encapsulated cells from host immune reaction [38,39].

In this review, we will focus on the cis-regulatory elements used in viral vectors. Choosing regulatory elements governing transcription is as important for the success of gene therapy as choosing the transgene and the type of the vector. It is considered that an AAV or LV expression cassette must contain ITR (inverted terminal repeats)/LTR (long terminal repeats), the promoter, and the polyadenylation signal (PAS) to be expressed. Other elements, such as enhancers, introns, or woodchuck hepatitis virus posttranscriptional regulatory elements (WPRE), are optional [40]. However, it was shown that PAS is not strictly necessary, at least in LVs [41]. At least one study used WPRE in place of PAS (in plasmids or in constructs inserted into the genome), and transgene expression was possible [42]. A diagram of obligatory and optional expression vector elements is shown in Figure 1.

Usually, specific promoters are used for targeting specific types of neurons. However, the actual picture in vivo is more complicated; individual genes are expressed in a variety of cell types in the brain, and so promoters are not specific to a single neuronal cell type. It is estimated that in the human genome, the number of cis-regulatory elements such as enhancers and repressors is at least an order of magnitude greater than the number of promoters [43].

### 1.2. Definitions of Promoters and Enhancers and Their Typical Properties

In eukaryotes, a promoter is a DNA region located nearby the transcription start site (TSS) of a gene and recognized by transcription factors (TFs). The binding of TFs with a promoter attracts RNA polymerase and starts the transcription process [44]. Promoters are usually located within 1000 bp from the TSS [45]. Promoter activation at a specific time and/or in a specific cell type or a specific state is often dependent on its interactions with other genome regions, including TSS-distal enhancers [44].

Most promoters described in this review are RNA polymerase II (pol II) promoters, because pol II transcribes mRNAs for protein-coding genes. The core (minimal) pol II promoter is typically about 80 bp long, from −40 to +40 relative to the TSS. Some of the known core promoter elements are the TATA box, the TFIIB recognition elements (BREu and BREd), the downstream core element (DCE), the initiator (Inr), the TCT motif, the motif ten element (MTE) and the downstream promoter element (DPE). These elements interact directly with components of the basal transcription machinery. Most studies of this machinery have been performed with promoters containing the TATA box as an essential core element. Although core pol II promoters were originally thought to be invariant, they have been found to have considerable structural and functional diversity. Among eukaryotic gene promoters, there are examples lacking some or even all of the usual core promoter elements listed above. Different combinations of motifs confer specific functional properties to the core promoter, e.g., the ability to function in concert with specific enhancers [46,47].

The original definition of an enhancer was a DNA sequence that activates transcription independent of direction and distance to a promoter. The sources of enhancer activity are DNA motifs recognized and bound by TFs. Enhancers are often located kilobases away from their target gene. They are usually transcribed and give rise to enhancer RNA (eRNA) [48]. Enhancer transcription is important for maintaining an open chromatin state, accessible to TFs. The majority of eRNAs are unstable, but they were shown to facilitate target gene transcription via many different mechanisms (reviewed in [49]). The classical enhancer definition has become even more blurred as a result of the identification of putative enhancer elements using novel high-throughput techniques. Multiple “enhancer-like” elements that might facilitate target gene expression via unusual mechanisms have been described [48].

There are some differences in sequence and epigenetic properties between typical promoters and enhancers. In vertebrates, many promoters contain CpG-rich regions, while enhancers are more often CpG-poor. Promoters are generally associated with higher levels of trimethylation of lysine 4 at histone 3 (H3K4me3) compared to monomethylation of the same residue (H3K4me1), while the opposite is true for enhancers in a poised state [44]. However, about a quarter of all human promoters are non-CpG-related, and their upstream peak of H3K4me3 is significantly lower than that of CpG-related promoters; CpG- and non-CpG-related promoters also differ in some other histone markers [50]. On the other hand, increased H3K4me3 levels are observed in enhancers when eRNA is actively transcribed [49]. Both promoters and enhancers have high levels of acetylation of lysine 27 at histone 3 (H3K27ac) when activated [44].

The key difference is that a promoter contains the TSS. However, the same element may function as a promoter for one gene and as an enhancer for another gene because promoters and enhancers may have very similar structures. It was confirmed that promoters can evolve to become enhancers and vice versa [44]; 2–3% of promoters in human cell lines can also act as enhancers controlling the expression of proximal and distal target genes [51]. Enhancer elements may be located within the promoter of the same gene; one of the examples of this is the *GFAP* (glial fibrillary acidic protein) gene [52]. By one definition, even remote enhancers are considered a part of a promoter. In this case, the promoter region is divided into (1) the core promoter responsible for the binding of the transcription apparatus; (2) the proximal promoter containing regulatory elements and located up to a few hundred bp upstream of the TSS; and (3) the distal promoter containing remote enhancers and silencers that may be located kilobases upstream of the TSS [53].

In the context of designing gene therapy vectors, combined regulatory elements including the enhancer, the promoter, and the 5′-UTR of the source gene may be sometimes referred to together simply as “promoter” [54], even though some other researchers prefer to avoid using this term for complex elements where a promoter is combined with other sequences [14].

### 1.3. General Considerations for Promoter Design for Viral Vectors

Universal promoters have an advantage for use in gene therapy of diseases when multiple cell types need to be treated, including some neurogenetic pathologies. In this case, it is desirable to express the therapeutic transgene in as many cells as possible [55]. However, other diseases (including epilepsy) need a more targeted approach, and current efforts are more focused on the selection of tissue-specific and even cell type-specific promoter sequences.

One of the most important considerations when designing cell type-specific viral vectors is the size of the promoter. AAVs, the most commonly used viral vectors, have a relatively small packaging capacity of about 4.4 kb. An important advance made in the AAV vector technology is represented by the self-complementary vector (scAAV) that carries two complementary copies of the DNA insert linked through a mutated ITR. scAAVs have 10- to 100-fold higher transduction efficiency compared with single-stranded AAVs. However, their packaging capacity is cut in half, to approximately 2.2 kb of foreign DNA [54]. Additional regulatory elements and fluorescent reporters are usually added to the expression cassette. All this severely limits the size of the transcribed transgene and its promoter; even with single-stranded AAV vectors and smaller transgenes, the maximum size of the promoter can only reach about 2 kb, which is actually smaller than many naturally occurring cell type-specific promoters [52,56].

One strategy of designing artificial promoters involves identifying the minimal/core promoter sequence through testing different shortened sequences of natural gene promoters. Another strategy is to assemble hybrid promoters using elements from different known promoters and enhancers [57]. These two strategies may be combined [58]. Usually, vertebrate promoters used for artificial vectors are derived from human or mouse genes; however, genes of other species may be used as well [59]. Combining viral and vertebrate promoter elements is also possible [60,61].

## 2. Different Types of Promoters Used in Viral Expression Vectors

In Table 2, we have summarized the promoters and transgenes used in viral vectors in different studies related to gene therapy for epilepsy. This table does not include studies of general properties of promoters and viruses also described in the present review; there are only papers in which therapeutic transgenes were tested in epilepsy models. Most of these studies are described in detail in the text below.

### 2.1. Universal Promoters

There are multiple universal (also called ubiquitous, or constitutive) promoters used in expression vectors. Universal promoters are supposed to work in different tissues with similar efficiency. The most well-known examples are cytomegalovirus immediate-early promoter (CMV), simian virus 40 early promoter (SV40), chicken β-actin promoter coupled with CMV early enhancer (CAG), promoters of genes encoding human elongation factor 1α (EF1α), human β-actin (ACTB), ubiquitin C (UBC), β-glucuronidase (GUSB), and mouse phosphoglycerate kinase 1 (PGK) [40,107,108]. These promoters may be viral (like CMV), derived from ubiquitously expressed vertebrate genes (like EF1α), or hybrid (like CAG). Ubiquitous promoters used for designing AAV-based vectors may be as short as 30 bp (but this core CMV promoter is not active without enhancers) or as long as 2500 bp (EF1α) [109].

A comparison of eight mammalian cell types transduced with LV vectors carrying GFP reporter driven by different universal promoters showed that UBC and PGK are consistently weak, while EF1α and CAG promoters are consistently strong in all the cell types. The CMV promoter is the most variable, being very strong in some cell types and rather weak in others [108].

Since a part of a pol II promoter is transcribed and included in the 5′-UTR sequence, it may affect the mRNA properties (like secondary structure or localization in the cell) and its translation rate. It was found that among universal promoters commonly used in expression vectors, “stronger” promoters not only drive more effective transcription, but also favor increased translation efficiency. In a recent study, Peterman et al. compared CMV, CAG, EF1α, CMV, UBC, hPGK (human PGK) and EFS (a shortened derivative of EF1α promoter) for their transcription and translation rate. These promoters were inserted in plasmids expressing mRuby2 fluorescent reporter and transfected into HEK293T cells. The authors used hybridization chain reaction RNA-FISH (HCR Flow-FISH) combined with protein quantification during flow cytometry to simultaneously measure the mRuby2 protein level and its mRNA level in the same cells with single-cell resolution. Promoter strength as measured by protein expression, ranked from highest to lowest, was CAG > EF1α > CMV > UBC > EFS > hPGK, consistent with the data in the literature. mRNA expression was more variable than protein expression, but the relative ordering of promoter strength determined by mRNA expression mostly matched that of protein expression. Higher mRNA levels strongly correlated with higher protein levels; however, relative protein-to-RNA ratios differed, indicating differences in effective translation rate across promoters. In addition to having higher mRNA levels, strong promoters (CAG, EF1α, and CMV) exhibited more efficient translation than weak promoters (UBC, EFS, and hPGK). Notably, splicing within the 5′-UTR increased the effective translation rate: for EF1α, it was 6-fold higher than for EFS, and the only difference between their sequences is the intron present in EF1α [42].

In the context of epilepsy, Mora-Jimenez et al. compared three different universal promoters, CMV, CAG and EF1α, for their ability to express *SCN1A* cDNA from plasmids in HEK293 cells. Expression levels of SCN1A mRNA and protein were similar for CMV and CAG, but lower for EF1α. CAG and EF1α were then used for further experiments with *SCN1A* expression in vivo using HC-AdV vectors. It was demonstrated that HC-AdVs with CAG and EF1α promoters have similar kinetics of reporter expression when injected into the basal ganglia or cerebellum of mice, with a rapid decay during the first 4 weeks and then a prolonged stabilization stage. The CAG promoter that showed higher expression levels in vivo was then used for *SCN1A* expression in murine brain (this experiment is described in Section 2.1.3) [36].

#### 2.1.1. Cytomegalovirus Promoter and Its Activity in Neurons Depending on the Neuronal Activation

Human cytomegalovirus major immediate-early promoter/enhancer (CMV promoter, also called hCMV) is a powerful and versatile enhancer-promoter unit used for mammalian expression vectors [110,111,112,113]. This promoter is very well-studied, so nowadays it is often used as a control promoter with known properties in experiments wherein the expression efficiency and cell specificity of novel promoters are tested.

The CMV promoter is rate-limiting for cytomegalovirus infection; it contains five cAMP-response elements (CREs), which allow the virus to exploit cAMP-response element-binding protein (CREB)-dependent mechanisms of transcription stimulation in the host cell. The CMV promoter and enhancer also contain multiple binding sites for other cellular TFs, such as nuclear factor κB/Rel (NF-kB/Rel), specificity protein 1 (SP-1), ETS transcription factor (ELK-1), retinoic acid receptor family (RAR-RXR), and activator protein 1 (AP-1) [112,114].

In neurons, activity-dependent changes in gene expression involving CREB are necessary for plasticity. CREB-dependent processes are associated with learning and memory, pain, and drug addiction. There are conflicting data about ectopic gene expression in neurons under the control of the CMV promoter in different studies, but there are many indications that the activity of this promoter depends on neuronal activation. Experiments with cultured neurons have shown that the CMV promoter acts as a molecular switch in neurons, and is strongly induced by membrane depolarization and other stimuli related to neuronal activity [114,115]. In vivo experiments demonstrated that the expression of a luminescent reporter (*Gaussia* luciferase, GLuc) increases after the injection of amphetamine in the brain transduced with a virus containing the GLuc gene under either the CMV promoter, or the CMV promoter enhanced by the β-globin gene intron and the intron of an immunoglobulin gene heavy-chain variable region [114].

It is known that the CMV promoter in viral vectors is prone to silencing in a few weeks after the infection [54,116]. However, the addition of other regulatory elements, for example, introns, may prolong expression time for transgenes driven by the CMV promoter [117]. Viral expression silencing as an obstacle for gene therapy is discussed in detail below, in Section 3. Despite the silencing problem, the standard CMV promoter was used for developing vectors for the gene therapy of epilepsy.

The viral expression of secreted proteins like NPY [62], glial cell line-derived neurotrophic factor (GDNF) [63] and fibroblast growth factor 2 (FGF-2) [64] under the control of the CMV promoter had anti-epileptic or anti-excitotoxic effects in various animal models. The overexpression of NPY driven by the CMV promoter, with an AAV vector injected intracerebroventricularly (i.c.v.), prolonged seizure latency and decreased seizure severity up to 4 weeks after the injection in the rat model of kainic acid (KA)-induced epilepsy [62]. One study took advantage of the CMV promoter’s transient functionality. Paradiso et al. aimed to locally supplement neurotrophic factors in the hippocampus at the stage when neuronal loss associated with epileptogenic damage is already present. The authors used the CMV promoter to cause time-constrained FGF-2 overexpression promoting neurogenesis. At the next step, the neuronal differentiation was facilitated by BDNF (brain-derived neurotrophic factor) overexpression driven by the ICP0 promoter. Both transgenes with their corresponding promoters were packed in the same HSV-1 vector that was injected into the rat hippocampus 3 days after the induction of status epilepticus (SE) by pilocarpine (at this time point, neurodegeneration is already noticeable, but spontaneous seizures have not appeared yet). This combined approach decreased the frequency and severity of spontaneous seizures. Out of 11 animals in the experimental group, 2 were seizure-free during the 20-day observation period. It was also shown that the viral expression of these factors prevents epileptogenesis but not ictogenesis (seizure onset). If the virus was injected later (3 weeks after the first spontaneous seizure), it did not affect the frequency, duration, or severity of the seizures [64].

Immunofluorescence images obtained 3 weeks after the injection of the LV vector expressing potassium channel K_v_1.1 and GFP under CMV promoter showed the colocalization of GFP with glutamatergic neuron marker α-subunit of calcium/calmodulin-dependent protein kinase II (CaMKIIα), and little or no colocalization with either the GABAergic neuron marker GAD67 (glutamate decarboxylase 1, 67 kDa) or the glial marker GFAP [65].

An adenoviral vector expressing the light chain component of the clostridial tetanus toxin (a synaptic transmission inhibitor) driven by the CMV promoter decreased the duration of EEG discharges in a model of penicillin-induced seizures in rats when injected into the motor cortex [35,66].

In 2000, During et al. employed an unusual strategy to decrease the number of N-methyl-D-aspartate (NMDA) glutamate receptors and thus decrease neuronal excitability. The animals were immunized against the NR1 receptor subunit by an AAV oral vaccine expressing *NMDAR1* under the control of the CMV promoter. The transgene expression persisted for at least 5 months, and was associated with a robust humoral immune response. The vaccine administration caused strong anti-epileptic and neuroprotective effects in rats with kainate-induced seizures [67]. However, we could not find any further research on antiepileptic viral vaccines, and the therapeutic benefits of the antibody-mediated targeting of NMDA receptors remains controversial [118].

When designing artificial promoters, parts of the standard CMV promoter, the enhancer or the minimal promoter, may be used separately. Combinations of these parts with other regulatory elements would have different properties [40,109,119]. It should be noted that when the minimal CMV promoter is used as a core part of artificial promoters, gene expression localization may be different compared with CMV-driven expression. For example, Haberman et al. [68] used AAV vectors with either the standard CMV promoter or a tetracycline-dependent [120] promoter with a minimal CMV core to express antisense RNA for *NMDAR1* in the rat inferior collicular cortex, aiming to knock down NMDA receptor expression. When the CMV promoter was used (AAV-CMV-NR1A virus), the antisense RNA increased the threshold for seizure initiation by electrical stimulation, as expected. However, the same RNA when expressed under the Tet-Off promoter (AAV-tTAK-NR1A virus) had the opposite effects. The coinjection of vectors with these two promoters and different reporter genes (AAV-tTAK-GFP and AAV-CMV-LacZ) showed that these viruses transduced distinct neuronal populations with only partial overlap. This may explain the contradictory results with NR1A RNA expression, since differing transduction ratios of inhibitory interneurons to primary output neurons may cause divergent seizure influences [68]. Drug-dependent promoters are described in detail in Section 2.3.

The CMV enhancer may also be used separately to construct novel promoters. The most well-known example of a universal hybrid promoter with this enhancer is the CBA promoter described in Section 2.1.3.

It is also possible to combine the CMV enhancer with cell-specific promoters. Liu et al. designed a hybrid promoter by appending a 380 bp fragment of the CMV enhancer upstream of the neuron-specific PDGF-β promoter. It was used in a plasmid vector with luciferase reporter. In neuronal cell lines (PC12 and C17.2) and in the rat brain (after injections in striatum or hippocampus), the hybrid promoter showed considerably enhanced activity compared with standard PDGF-β. The neuronal specificity of the PDGF-β promoter was preserved in the hybrid construct [119]. Hioki et al. added the CMV enhancer to five different neuronal promoters, with varying results. Although the transgene expression driven by the hybrid promoters increased, the neuronal specificity decreased in most cases. The best result was obtained for the SYN promoter (described in Section 2.2.1) [61].

#### 2.1.2. Herpesvirus Promoters

HSV vectors are used for studying gene therapy for epilepsy in animals. While these viruses are modified and replication-defective, their native promoters may be kept intact and used for expressing transgenes in the CNS.

The intrahippocampal injection of an HSV vector expressing heat shock protein Hsp72 under control of the viral promoter α4 decreased neuronal death in the dentate gyrus caused by systemic KA administration to rats [69].

When KA was injected into an area dorsal to the apex of the dentate gyrus together with an HSV vector expressing Glut-1 glucose transporter under the viral promoter α22, KA-induced neurotoxicity was decreased. Moreover, the virus improved memory parameters of mice that received KA [70].

Interestingly, not only native HSV promoters, but unmodified viral genes could be used for treating epileptic seizures in animal models. The intranasal administration of the growth-compromised HSV-2 vector ΔRR prevented KA-induced seizures, neuronal loss and inflammation in rats and mice. The authors suggest that these effects were caused by the viral antiapoptotic gene ICP10PK (expressed under its own native promoter). Moreover, they established a connection between AP-1 complex activation and ICP10PK expression. This is particularly relevant for the use of this virus as a neuroprotective agent because neurotoxic stimuli upregulate AP-1. ICP10PK was expressed in the hippocampus of the ΔRR-treated animals for at least 42 days in the absence of virus replication and late virus gene expression [71].

For the HSV promoter ICP0, transgene expression duration for at least 14 days in culture was demonstrated [121]. In the paper by Paradiso et al. cited above, the authors used this promoter for the long-term expression of BDNF, while the CMV promoter was used for transient FGF expression. Injection of the replication-defective HSV with these insertions decreased signs of epilepsy in the rat pilocarpine model [64].

#### 2.1.3. CBA (CAG) Hybrid Promoter and Its Derivatives

CBA is an artificial hybrid promoter containing CMV immediate early enhancer and the promoter of the chicken β-actin gene. However, other elements (usually other parts of the chicken β-actin gene or a splice acceptor of the rabbit β-globin gene [60,122]) may be added to this promoter, and in the literature there are controversial data about length, elements and names for CBA and its derivatives (CAG, CAGG, CAGGS). Sometimes, in different studies, the same name is used for promoters with different compositions, which may be confusing. In this section, we are going to use the names as provided in the referenced papers.

Mora-Jimenez et al. chose the CAG promoter for the expression of *SCN1A* in an HC-AdV vector in a mouse model of DS. The HCA-CAG-SCN1A vector was injected into basal ganglia of *Scn1a*^WT/A1783V^ mice that have a severe DS phenotype. One week after surgery, electrophysiological recordings were performed, and a dose-dependent reduction in interictal epileptiform discharges was observed in mice treated with the virus. To test the influence of the virus on survival and seizure susceptibility, the HCA-CAG-SCN1A vector was administered to 5-week-old *Scn1a*^WT/A1783V^ mice in two or three brain regions (basal ganglia and cerebellum, also prefrontal cortex for some animals). A significant decrease in mortality was observed in mice treated with HCA-CAG-SCN1A by both administration methods. One month after treatment, mice were subjected to hyperthermia to determine their seizure threshold temperature. Only mice that had HCA-CAG-SCN1A injected in three brain regions showed a significant increase in the threshold (compared with control mice or their own pre-treatment values). Mice that received a triple injection of the virus were then subjected to motor and behavioral tests. It was shown that HCA-CAG-SCN1A ameliorates motor and some behavioral signs of DS, but does not rescue hyperactivity or learning delay [36].

Natarajan et al. used the CBa promoter in AAV5 vectors to overexpress preprosomatostatin in rats kindled to seizures by repeated electric stimulation. The virus was injected bilaterally into the hippocampal dentate gyrus and CA1 region. After 3 weeks, rats that received the control AAV5-CBa-eGFP virus continued to exhibit high-grade seizures, while 6/13 rats that received AAV5-CBa-preproSST-eGFP were seizure-free [22]. Later, the same research group estimated the effects of the same virus (AAV5-CBa-preproSST-eGFP) on histological parameters of kindled rats—hippocampal neurogenesis and microglia activation. Both neurogenesis and neuroinflammation are known to be potentiated as a result of kindling. It was found that rats that responded to the virus treatment (as estimated by decrease in seizure parameters) also had significantly fewer dividing Type-1 neural stem cells (NSCs) in the hippocampi, suggesting that sustained SST expression exerts antiepileptic effects through the normalization of neurogenesis. The viral treatment did not revert kindling-induced microglia activation in the hippocampus [72].

Noé et al. induced AAV1/2-mediated NPY overexpression in rat hippocampus by use of the CBA promoter. Radioimmunoassay measurements demonstrated an average 10-fold increase in NPY levels. This overexpression lasted for at least 6 months. Brain tissue close to the AAV vector injection site did not show neuronal cell loss, or microglia or astroglia activation. The pattern of the target gene’s overexpression was very similar to the one observed using the neuron-specific NSE promoter. The efficacy of this approach in reducing spontaneous seizures in epileptic rats was highly dependent on the extent of NPY increase in mossy fibers and in fibers of the outer molecular layer of the dentate gyrus and hippocampus proper [23]. More recently, Melin et al. used the CAG promoter and the internal ribosome entry site (IRES) to drive the expression of bicistronic combinations NPY-IRES-Y2 or Y2-IRES-NPY in different AAV serotypes in rats. Acute seizures were induced by KA three weeks after the intrahippocampal injection of the viruses, and seizure parameters were assessed. The AAV1-NPY-IRES-Y2 vector had the most prominent anti-seizure effects and was evaluated further in resected brain tissue from patients with drug-resistant TLE. AAV1-NPY-IRES-Y2 increased the expression of NPY and its receptor Y2R, and caused a decrease in glutamate release from excitatory neuron terminals in human hippocampal tissue [74].

Another neuropeptide, dynorphin, was also expressed under the control of the CBA promoter to prevent seizures. The preprodynorphin gene with CBA promoter was packed into self-complementary AAV2-based vector backbones and AAV serotype 1 capsids. Viruses were administered into the epileptogenic focus in mice 1 month after the KA injection, or into the dorsal hippocampus in rats in the electroconvulsive model. Viral injections provided a quickly established seizure-preventing effect in both mouse and rat models of TLE lasting up to 3 months. The neuron-specific expression of the transgene in the hippocampus was observed for 5.5 months after the injection. It is supposed that both excitatory and inhibitory neurons were transduced [20].

The 1.6-kb hybrid CAGGS promoter composed of the CMV immediate-early enhancer, CBA promoter, and CBA intron 1/exon 1 has been shown to provide ubiquitous and long-term expression in the brain [54,60,123]. An 800-bp miniature version of the CAGGS promoter was made by replacing the CBA 5′-UTR with a truncated simian virus 40 (SV40) intron. The smaller size of this promoter proved especially useful for scAAV vectors. Gray et al. created the CBh promoter, a novel hybrid form of the miniature CBA promoter described earlier by other researchers. CBh is also 800 bp long, and it was made by replacing the SV40 intron in the miniature CBA by a hybrid intron composed of a 5′-donor splice site from the chicken β-actin 5′-UTR and a 3′-acceptor splice site from the minute virus of mice (MVM) VP gene intron. For testing the promoter activity, scAAV9 vectors with GFP reporter were used. CMV and miniature CBA promoters served as controls to compare with CBh. Expression levels driven by different promoters in the CNS were tested ex vivo in rat hippocampal slices and in vivo in adult mice after intravenous (i.v.) or intrathecal injections. It was established that the CBh promoter provided strong, long-term (up to 10 weeks), and ubiquitous transgene expression in all observed types of CNS cells. CBh has advantages compared to both CMV (which initially drives strong expression but is silenced later) and miniature CBA (less universal because it drives much weaker expression in motor neurons of the brainstem and spinal cord) [54].

Hordeaux et al. investigated the long-term performance of AAV9 virus expressing human *IDUA* gene under a promoter that is called CB7 in this study, but described as a version of CBA: a hybrid promoter containing the CMV enhancer coupled with the chicken β-actin promoter [79]. This paper is described in detail in Section 3.

#### 2.1.4. Elongation Factor 1α Promoter

Translation elongation factor 1α (EF1α) is expressed in multiple cell types [124]. As was mentioned above, the comparison of multiple promoters in different cell types showed that the EF1α-driven expression level is similar to that of the CAGG promoter [108].

The ability of this promoter to work in different cells including GABAergic neurons was used in the study wherein a closed-loop system for decreasing seizure duration was established. Hristova et al. injected the AAV-EF1a-DIO-hChR2(H134R)-mCherry-WPRE-pA virus into the medial septum of VGAT-IRES-Cre transgenic mice. The seizures were induced by the intrahippocampal administration of KA. The researchers implanted a micro-LED near the medial septum for the optogenetic stimulation of GABAergic inhibitory neurons. To register seizures, local field potential (LFP) electrodes were implanted in different parts of the hippocampus. These elements were connected in such a way that at the beginning of a seizure the LED would turn on and emit light at the septal inhibitory neurons, which causes their activation via the channelrhodopsin-2 expressed by the virus and decreases seizure duration [80].

We summarized the pros and cons of the three universal promoters most commonly used for transgene expression in the brain in Table 3.

To conclude this section, universal promoters in gene therapy of epilepsy may be used in cases when it is necessary to transduce as many neurons as possible (like when the chosen transgene is a normal allele replacing the mutated one, or an inhibitory neuropeptide). The preferential targeting of neurons may be achieved by a careful choice of the virus serotype or administration route. However, a possible early silencing for exogenous promoters like CMV should be considered.

### 2.2. Cell Type-Specific Promoters Relevant for Epilepsy Treatment

There are a few commonly used promoters that are considered neuron-specific, but which drive gene expression in multiple types of neurons indiscriminately; these include promoters of genes encoding synapsin I (SYN), CaMKIIα, tubulin α1 (Ta1), neuron-specific enolase (NSE) and platelet-derived growth factor β-chain (PDGF) [61]. The *GFAP* gene promoter and its shortened forms may be used to restrict expression to astrocytes [52].

In this section, we will describe some of these promoters, as well as some more specific promoters that are only functional in certain types of neurons and may be useful in the context of epilepsy studies and/or gene therapy.

#### 2.2.1. Synapsin I Promoter

Synapsins (encoded by three different genes, each producing a few splice isoforms) are the first discovered presynaptic proteins. The best-known function of these proteins is to regulate synaptic transmission by controlling the storage and mobilization of synaptic vesicles within a reserve pool. However, the loss of synapsins differentially affects transmission in different neuron types [129]. Synapsin I represents approximately 6% of the total protein in the highly purified synaptic vesicle fraction [130]. In 2003, Kügler et al. demonstrated that a small fragment of the human synapsin 1 gene promoter (about 470 bp) is able to restrict transgene expression from an adenoviral vector exclusively to neurons. This promoter showed more neuronal specificity than the NSE promoter, which also supported reporter expression in NeuN-negative cells [34]. In different papers, this promoter, when used in expression vectors, may be called SYN, hSyn, hSyn1 or SynI.

Hioki et al. generated lentiviral vectors with five different neuron-specific promoters with or without fused CMV enhancer, and then characterized their neuronal specificity and transcriptional activity in rat neostriatum, thalamic nuclei and cerebral cortex, using GFP as a reporter. The CMV promoter was used as control (GFP expression driven by CMV promoter was observed in both neuronal and glial cells). GFP production by enhanced promoters was about 2–4 times higher than without the enhancer, but neuronal specificity was significantly decreased in most cases. SYN promoter displayed the highest specificity for neuronal expression in all the regions examined (more than 96%). In the case of SYN promoter, adding the enhancer (E) caused the least decrease in specificity compared to other studied promoters. The authors also estimated the time course of transcriptional activity and neuronal specificity from 3 days to 8 weeks after the injection of LV vectors with SYN and E/SYN promoters into the neostriatum. GFP fluorescence linearly increased even up to 8 weeks for both promoters, indicating that their transcriptional activity can be maintained at least that long. Furthermore, the neuronal specificity of both promoters seemed constant throughout 8 weeks [61].

One of the methods proposed for epilepsy treatment is to block the expression of GluK2, a kainate receptor subunit encoded by the *Grik2* gene. This receptor is involved in recurrent axons sprouting in the hippocampal dentate gyrus during epilepsy. The sprouting leads to abnormal neuronal activity in the hippocampus, and epilepsy. Boileau et al. induced GluK2 knockdown in neurons using neuron-specific miRNA expression under the SYN promoter. The transduction of rat hippocampal cultures with an AAV9 vector expressing this miRNA reduced total GluK2/GluK3 protein levels by 35% (as compared to AAV9-GFP). In vivo, the AAV9-hSyn1-miGRIK2 transduction in dentate gyrus decreased seizure events in a pilocarpine model of epilepsy. At the present time, the AAV9-hSyn1-miGRIK2 viral construction is being investigated in clinical trials as a treatment for unilateral refractory mesial TLE [13,81].

Another method is a chemogenetic approach to silence neurons during seizures. Modified muscarinic receptor hM4Di decreases neuronal excitability by activating Gα_i_. Miyakawa et al. injected the virus AAV2.1-hSyn-FLAG-hM4Di-IRES-AcGFP into the primary motor cortex of two monkeys, then injected a GABA_A_ receptor antagonist bicuculline into the same region and observed seizures with electrocorticography (ECoG) and video registration. hM4Di agonist injection resulted in a reduction in seizure amplitude, the disappearance of multi-wave complexes and a reduction in clonic seizures [82].

#### 2.2.2. Neuron-Specific Enolase Promoter

Neuron-specific enolase (NSE) is a glycolytic enzyme expressed in most terminally differentiated neurons and neuroendocrine cells, but not in other cell types (including neuroblasts). NSE is one of the most abundant brain-specific proteins. The ability of the *NSE* gene promoter region to target the expression of a reporter gene to postmitotic neurons was first established in 1990 with transgenic mice expressing β-galactosidase under control of this promoter [131].

In one of the early experiments with gene overexpression in the brain, Navarro et al. used the rat NSE promoter in an adenoviral vector (Ad-NSE) to target gene expression to neurons. A similar vector with Rous sarcoma virus LTR promoter (Ad-RSV) was used as control. Both adenoviruses were injected into rat hippocampus, cerebellum and striatum. Ad-NSE preferentially transduced neurons and had less cytotoxicity than Ad-RSV. The marker gene expression after Ad-NSE infection remained stable for 3.5 months, and was detectable for 6 months [33].

In 2002, Klein et al. were the first to evaluate the dose response of the NSE promoter in transduction experiments in vivo using AAV2 vectors and GFP reporter. The number of GFP+ cells was estimated 2 months after the virus injection into rat medial septum. The calculated ED_50_ value was 2.59 × 10^9^ particles with *B*_max_ (the maximal number of binding sites) of 601 cells for the vector with the GFP coding sequence directly following the promoter; for the bicistronic vector with an intervening transgene (myc-tagged BDNF) and an IRES element between the promoter and the GFP coding sequence, ED_50_ was higher, at 8.76 × 10^9^ particles with *B*_max_ of 662 cells. The intervening sequence in the bicistronic vector resulted in fewer GFP+ cells at low virus doses, but similar numbers of GFP+ cells at the highest dose tested [123].

Xu et al. compared multiple universal and cell type-specific promoters for their ability to drive transgene expression in primary neuronal cultures and in the brain when used in AAV2 vectors. It was shown that NSE (1.8 kb) drives stronger expression than the shortened NSE version (0.3 kb). In descending order, for all tested promoters, their relative activities in vitro were as follows: NSE 1.8 kb > EF > NSE 0.3 kb ≈ GFAP > CMV > hENK > PPE ≈ NFL ≈ NFH > nAchR. In vivo (the viruses were injected into different regions of rat brain), the NSE 1.8 kb promoter showed the highest reporter expression levels, with the ranking of promoter activities being NSE 1.8 kb + WPRE > NSE 1.8 kb > EF > GFAP > NSE 0.3 kb > PPE > hENK > CMV > NF ≈ nAchR > NFH in the hippocampus [132].

The NSE promoter is widely used to express genes of neuropeptides like galanin [83] and NPY [84,85], as well as the NPY receptors Y2 and Y5 specifically in neurons. These are “anti-epileptic” NPY receptors, while its other cognate receptor, Y1, is “pro-epileptic” [24]; this is why when the viral expression of NPY is combined with the expression of Y2 [86,87] or Y5 [88], the anti-seizure effects are more prominent as compared to the expression of NPY alone.

In a study where the effect of AAV-induced long-lasting NPY overexpression in the hippocampus in different models of epilepsy in rats was investigated, the expression pattern of NPY under the NSE promoter was AAV serotype-dependent. The AAV2 vector increased NPY expression in hilar interneurons only (a population of cells that normally express NPY in physiological conditions), whereas the chimeric serotype 1/2 vector caused more widespread expression, including ectopic expression in mossy fibers and in pyramidal cells and the subiculum. For the serotype 1/2, expression in dentate gyrus and CA1 was higher than for serotype 2. EEG seizures induced by intrahippocampal kainate were reduced by 50–75%, depending on the vector serotype, and seizure onset was markedly delayed. 1/2 rAAV–NSE–NPY treatment resulted in the abolition of SE in rats, and also effectively delayed kindling epileptogenesis [84].

The influence of NPY on seizures was also studied in the Genetic Absence Epileptic Rats from Strasbourg (GAERS) model of absence epilepsy. NPY overexpression in an AAV vector under the NSE promoter in the thalamus and somatosensory cortex reduced the number of seizures. The duration of seizures was reduced when the virus was injected into the thalamus but not into the cortex [85].

However, some researchers consider the pan-neuronal promoter NSE to be less suitable for the expression of NPY and Y2 than the minimal CaMKIIα (0.4 kb; mCamk2a) promoter, which has more specificity for excitatory neurons. In the case of mCamk2a, less inhibition of inhibitory neurons is expected [94].

Fadila et al. tested an unusual viral vector, CAV-2, for its ability to deliver the *SCN1A* transgene in a model of DS. The authors first tried a number of commonly used promoters with this virus and chose NSE as the most suitable one. After intrahippocampal injection, CAV-2 with NSE promoter supported the robust neuronal expression of the reporter in the hippocampus and the less prominent reporter expression in the glutamatergic neocortical hippocampal projecting neurons. CAV-SCN1A or CAV-GFP (control) viruses with this promoter were injected in the hippocampus of 5-week-old wild-type (WT) or *Scn1a*^A1783V/WT^ mice. In the mutant mice, CAV-SCN1A reduced the occurrence of epileptic spikes (assessed by ECoG), and reduced susceptibility and elevated the temperature threshold for heat-induced seizures. If CAV-SCN1A was injected earlier, at the onset of the symptoms (P21–P24), it reduced mortality by approximately 40% (compared with mutant mice injected with CAV-GFP). At 36 h after injection, short-term video recordings showed a reduction in the number of convulsive seizures in CAV-SCN1A-treated juvenile mutant mice. CAV-SCN1A injections in juvenile *Scn1a*^A1783V/WT^ mice either prevented thermally induced seizures, or increased the seizure threshold temperature and reduced the number of epileptic spikes in both cortex and hippocampus. Additional experiments showed that CAV-SCN1A rescued DS-associated network dysfunctions, restored background ECoG activity, and partially corrected behavioral deficits in *Scn1a*^A1783V/WT^ mice. When the virus was injected in the thalamus, it had similar anti-epileptic effects. Finally, CAV-SCN1A was tested in another DS mouse model (*Scn1a*^R613X/WT^). Similarly to *Scn1a*^A1783V/WT^ mice, CAV-SCN1A injections improved survival and reduced susceptibility to thermally induced seizures in *Scn1a*^R613X/WT^ mice [37].

#### 2.2.3. Promoters with Suggested Specificity for Principal Neurons: CaMKII and VGLUT1

For a long time, it was assumed that in the cortex, only excitatory principal neurons, but not inhibitory cells, express calcium/calmodulin-dependent protein kinase II subunit α encoded by the *Camk2a* gene. This relies on the results obtained by immunohistochemical studies in the 1990s, showing that CaMKIIα is present in cortical glutamatergic cells, but not in GABAergic interneurons [133,134]. This, and the subsequent report of transgene expression being restricted to cortical pyramidal neurons that was achieved with the mouse CaMKIIα promoter in an LV vector [135], led to a widespread use of this promoter for targeting principal cells. In different papers, this promoter in expression vectors may be called CaMKIIα, CaMKIIa, α-CaMKII, mαCaMKII, Camk2a or CaMKII.

However, later, it became obvious that in some cases, this promoter can also be active in inhibitory interneurons. Nathanson et al. observed that the same promoter as in [135], when used in an AAV vector, had a bias towards gene expression in excitatory cortical neurons, but did not drive gene expression exclusively in these cells [136]. Schoenenberger et al. also reported the expression of a transgene delivered in an AAV virus and driven by the CaMKII promoter in SST+ and PV+ interneurons in the hippocampus [137]. Recently Veres et al. demonstrated that other types of interneurons are targeted by AAV vectors with the CaMKII promoter as well. The reporter proteins were detected in at least 5 types of GABAergic interneurons (PV+, SST+, neuronal nitric oxide synthase (nNOS)+, NPY+, and cholecystokinin (CCK)+ neurons) [138]. The CaMKII promoter showed even less cell type specificity in cultured neurons. Egashira et al. tested different promoters (in LV vectors) for their specificity for glutamatergic neurons in culture. In these experiments, the CaMKII promoter caused reporter expression not only in excitatory neurons, but also in ∼90% of GABAergic neurons. Moreover, the CaMKII promoter also worked in some glial cells [139]. All these results challenge the use of CaMKII promoter-driven protein expression as a selective tool for targeting cortical glutamatergic neurons using viral vectors. However, it may be possible that CaMKII (and some other promoters) have less strict cell type specificity in rodents compared to primates. In one study, SYN and CaMKII promoters showed excitatory neuron-specific expression in the monkey brain (SYN, 99.7%; CaMKII, 100.0%), but their specificities for excitatory neurons were significantly lower in the rat brain (SYN, 94.6%; CaMKII, 93.7%) based on NeuN and GABA staining [125].

It also must be noted that an exogenous CaMKII promoter may be inactive in certain subsets of cortical excitatory neurons. Wang et al. generated transgenic mice with the GFP reporter fused with the CaMKII promoter inserted into the genome; some neurons that expressed endogenous CaMKII (registered by immunostaining) did not coexpress GFP, suggesting that the CaMKII promoter in the insertion was not fully functional [140].

A few different therapeutic transgenes were tested with the CaMKII promoter in animal models of epilepsy. One of the most promising transgenes for treating epilepsy is the engineered potassium channel (EKC), a mutant version of *KCNA1*.

In a recent study, Almacellas Barbanoj et al. used the CaMKII promoter to express EKC in an AAV9 vector used to treat frontal lobe focal cortical dysplasia in a mouse model (in utero electroporation of frontal lobe NPCs with a constitutively active human Ras homolog enriched in brain (RHEB) plasmid). Mice in this model show epileptiform ECoG activity, spontaneous generalized seizures and behavioral abnormalities. The injection of AAV9-CAMK2A-EKC in the dysplastic region caused a robust decrease in the frequency of seizures (∼64% in relation to the baseline in epileptic animals). However, it did not rescue behavioral abnormalities. Also, AAV9-CAMK2A-EKC had no effect on interictal discharges or behavior in RHEB-transfected mice without generalized seizures [89].

Snowball et al. showed that EKC expression under the control of the CaMKII promoter in a nonintegrating LV vector decreased seizure frequency in a model of focal neocortical epilepsy in rats [15].

The same research group earlier demonstrated the anti-epileptic effect of halorhodopsin from *Natronomonas pharaonis* (NpHR) expressed under the control of the CaMKII promoter in an LV vector injected into the rat motor cortex. Seizure activity caused by tetanus toxin coinjected with the virus decreased after the optical stimulation [65].

However, the optogenetic approach requires lasers or LEDs being introduced directly in the brain, which limits its possible clinical application. This is why chemogenetics may be a better alternative. Kätzel et al. used a designer drug receptor hM4Di and its agonist clozapine-N-oxide. The virus AAV5-CaMKIIa-HA-hM4D(Gi)-IRES-mCitrine was injected into the primary motor cortex of rats. The systemic administration of clozapine-N-oxide suppressed focal seizures induced by pilocarpine and picrotoxin [90].

Recently, Nikitin et al. demonstrated that the Ca^2+^-gated potassium channel K_Ca_3.1 (encoded by *KCNN4* gene) may reduce ictal activity in brain slices when expressed from an AAV vector with the CaMKII promoter. The authors used the 4-aminopyridine in vitro model of epilepsy. It was shown that the inhibition of ictal activity was not accompanied by lower-frequency coding ability or changes in action potential shape [91].

The CaMKII promoter was also used as a component of systems aimed to increase the expression of endogenous genes with the CRISPRa approach [28,92]. The CRISPRa method is described in detail in Section 2.5.

Very recently, Cattaneo et al. used a shortened version of the CaMKII promoter (mCamk2a, 0.4 kb) to express NPY and its receptor Y2R in an LV vector for treating epilepsy in the synapsin triple knockout model in mice. The injection of the virus into dentate gyrus decreased the number of seizures and increased the latency of the first seizure (as counted from the start of observation). The authors also expressed the EGFP reporter in a similar vector with the mCamk2a promoter, and counted EGFP+ inhibitory interneurons in different parts of the hippocampus using immunostaining for different interneuron markers. The greatest number of interneurons among all EGFP+ cells was observed in the CA3 region—9.21% of EGFP+ cells there were also GAD1+, and 11.11% were PV+. EGFP expression was negligible in SST+ and NPY+ GABAergic neurons [94].

The reports about the CaMKII promoter being not specific for excitatory neurons (at least in rodents) mean that the search for specific promoters with these properties should be continued. One of the known candidates is the vesicular glutamate transporter 1 (VGLUT1) promoter. The *VGLUT1* gene encodes a protein that transports glutamate into synaptic vesicles. However, not all glutamatergic neurons express VGLUT1 because the VGLUT family includes three proteins. VGLUT1 and VGLUT2 are both expressed in glutamatergic neurons but exhibit a complementary pattern of expression in the adult brain. The cerebral cortex, hippocampus and cerebellar cortex express predominantly VGLUT1, while the brainstem and deep cerebellar nuclei almost exclusively express VGLUT2. Notably, VGLUT3 is expressed by non-glutamatergic neurons and even in some non-brain organs [141].

Zhang and Geller showed that the *VGLUT1* promoter region supports expression in vivo mostly in VGLUT1+, but not in VGLUT2+ glutamatergic neurons. The authors used HSV-1 vectors with LacZ as a reporter gene driven by either the *VGLUT1* promoter (pVGLUT1lac virus) or the control chimeric neuron-specific promoter. The viruses were injected into the rat brain stereotaxically, either in the postrhinal cortex (POR, contains mostly VGLUT1+ neurons) or in the ventral medial hypothalamus (VMH, mostly VGLUT2+ cells). Immunohistochemistry showed that the control virus supported β-gal expression in similar numbers of cells in the POR cortex and the VMH. In contrast, pVGLUT1lac supported expression in approximately tenfold more cells in the POR cortex than in the VMH. In the POR cortex, more than 90% of β-gal+ cells were VGLUT1+. Out of the few β-gal+ cells in the VMH, about 10% were VGLUT1+. However, the estimated fraction of VGLUT1-containing neurons in this region is much lower, so the virus was still preferentially expressed in these cells [142].

The VGLUT1 promoter used in the aforementioned study is a long genome region that contains both the upstream promoter (7 kb) and the first intron of the *VGLUT1* gene (4.6 kb). The same authors then examined it more thoroughly to identify shorter sequences responsible for cell type-specific expression. HSV-1 viruses with LacZ reporter and different parts of this long promoter were injected into the postrhinal cortex of rats. Immunostaining showed that the virus containing the whole long VGLUT1 promoter is effective for driving reporter expression preferentially in glutamatergic POR neurons (known to contain VGLUT1), but a minor fraction of GABAergic neurons was also expressing the reporter. The basal *VGLUT1* promoter (323 bp fragment that contains the TSS) was not cell type-specific, and supported the expression in glutamatergic and GABAergic neurons with similar efficiency. Different combinations of the basal promoter and parts of the long promoter region were then tested. It was found that when either the 7 kb upstream promoter region or the 4.6 kb first intron were fused to the basal promoter in an HSV-1 virus, the reporter expression pattern was similar to the whole long promoter—predominantly glutamatergic neuron-specific expression, with a small fraction of GABAergic neurons being labeled [143].

Egashira et al. tested several putative promoter sequences (SYN, CaMKII, VGLUT1, Dock10 and Prox1) for their selectivity for glutamatergic neurons. The experiments were performed on primary hippocampal cultures prepared from *VGAT-Venus* transgenic mice (which express Venus fluorescent protein in GABAergic neurons, so these are easily discernible). A pair of LV vectors was used for each promoter: one virus expressed advanced tetracycline transactivator (tTAad) under a tested promoter, the other virus expressed the TagRFP reporter driven by tetracycline response element (activated by tTAad protein). The results indicate that among the tested promoters, the 2.1 kb VGLUT1 promoter conferred the highest preference for glutamatergic neurons over GABAergic ones. It achieved highly preferential, but not exclusive, gene expression in glutamatergic cells. Immunostaining for GFAP confirmed that the VGLUT1 promoter does not drive any gene expression in glial cells. The analysis of various lengths (0.8–4 kb) of the *VGLUT1* gene promoter regions was performed to identify fragments that may be responsible for the glutamatergic selectivity, and a segment between −2.1 kb and −1.4 kb from the TSS was identified as such a region. However, this segment by itself (without the 1.4 kb sequence between it and the TSS) was not tested. Importantly, in this study, it was also found that the VGLUT1 promoter showed preference for the excitatory neurons expressing VGLUT1, but not VGLUT2. It was demonstrated using mixed neuronal cultures of the hippocampus and brainstem, in which excitatory neurons predominantly express VGLUT1 and VGLUT2, respectively. Both transporters were visualized by immunostaining. To identify transgene expression at presynaptic terminals, neurons were transduced with LV vectors expressing sypHy (synaptophysin fused with fluorescent protein) under the control of tested promoters. The VGLUT1 promoter mostly induced sypHy expression in VGLUT1+ presynaptic boutons, even though some rare VGLUT2+ ones also expressed the reporter. The SYN promoter used as control was able to drive sypHy expression at VGLUT1+, VGLUT2+ and both-negative boutons [139].

#### 2.2.4. Promoters Specific for GABAergic Interneurons and Other Strategies to Target These Neurons

For treating some kinds of epilepsy, targeting cortical GABAergic interneurons may be particularly useful. There are multiple subpopulations of interneurons—four or five major types and between 30 and 120 subtypes, depending on the criteria used [59,144,145,146]. The picture is even more complicated because even major types partially overlap in their gene expression patterns, and it differs by species. For example, in rats, both CCK+ and calretinin (CR)+ interneuron types overlap with vasoactive intestinal peptide (VIP)+, but VIP+ does not overlap with PV+ or SST+. In mice, unlike rats, many CR+ cells are also SST+ [59,147,148]. Very few mammalian promoters have been characterized well enough to use them for viral transgene expression specific to interneurons and interneuron types [149].

In a screening study, Nathanson et al. compared the efficacy and specificity of dozens of short promoters that may potentially target different interneuron types. The authors used promoters of some genes characteristically expressed in these types in mammals, their orthologs in fugu, and synthetic promoters in AAV1/2 or LV vectors. The viruses were injected into the mouse somatosensory cortex and thalamus, the rat somatosensory cortex and the monkey motor and somatosensory cortex. Unfortunately, none of the promoters tested were capable of restricting expression to specific inhibitory neuron types. However, four promoters (fugu NPY, fugu SST, mouse A93, E1.1-NRSE) preferentially drove expression in inhibitory neurons with minimal excitatory neuron expression. AAV-E1.1-NRSE, a construct with an artificial promoter, largely restricted expression to non-PV interneurons in mice [59].

Mantoan Ritter et al. tested the ability of fragments of the *GAD67* (also called *GAD1*) gene promoter to target all GABAergic interneurons, and the ability of the *CCK* gene promoter to target CCK+ interneurons. The *GAD1* gene encodes glutamate decarboxylase 1, which weighs 67 kDa, and is sometimes called GAD67 (especially when used as a marker of GABAergic cells). The authors generated LV vectors encoding opsin proteins driven by GAD1 or CCK promoters, and used these constructs to transduce neuronal cultures (prepared from the medial ganglionic eminence) and rat brains in vivo (intrahippocampal injection). In cultures, 90% of the neurons expressing the reporter from the CCK promoter-containing LV vector were CCK+. In vivo, the results were very different. Several GAD1/CCK promoter-containing constructs were expressed in vivo, with different efficiencies, but the expression was not specific for interneurons. For example, in CA1 and CA3, only 7.5% of cells expressing LV-GAD67-ChR2-mCherry were GABA+. This work underlines an important difference between in vitro and in vivo models. LV vectors with the CCK promoter may be useful to target CCK+ cells in cultures enriched in interneurons, but not in vivo [149].

Niibori et al. performed the overexpression of the Na_v_β1 sodium channel subunit in the *Scn1a^+/−^* mouse model of DS. They used an AAV vector with the truncated promoter of the mouse *Gad1* gene that is expressed in GABAergic neurons. The virus was injected i.c.v. and into cisterna magna (i.c.m.) of mouse pups at P2. Brain samples were collected for immunohistochemistry at the ages of P17–18 and 2–3 months. Mortality was monitored from P20 to P56 and spontaneous seizures from P20 to P30, and different behavioral tests were performed at the age of 8–11 weeks. More than 80% of all transfected cells were neurons (NeuN+). By subtracting the percentage of GAD-67/GABA-labeled cells from the total NeuN-labeled population, it was estimated that 50–70% of transgene-labeled cells in the immature brain and 75–85% in the mature brain were in fact non-GABAergic neurons, most likely pyramidal cells and other glutamatergic neurons. However, the virus injection alleviated the symptoms of DS. Untreated female *Scn1a^+/−^* mice have a higher degree of mortality than males, while males of this genotype show more prominent behavioral abnormalities. Treatment with the AAV-NaVβ1 virus had a prosurvival effect in *Scn1a^+/−^* mice, which was more apparent in females compared to males. *Scn1a^+/−^* males, but not females, showed significant reductions in spontaneous seizures and the normalization of motor activity and performance in elevated plus maze as a result of viral treatment. In GABAergic interneurons of *Scn1a^+/−^* mice, the overexpression of Na_v_β1 should potentiate residual Na_V_1.1 sodium channels. But it is also known that Na_v_β1 induces variable effects (potentiation or inhibition) on several potassium channels, and the overexpression of Na_v_β1 may suppress excitatory neuron firing, possibly by effects on K_v_4.2. The authors suggest that both these mechanisms contributed to the observed therapeutic effects of the virus [95].

Later, Tanenhaus et al. tested the efficacy of the human *GAD1* gene promoter to selectively target GABAergic inhibitory interneurons. The authors worked with a genetic mouse model of DS. To increase *SCN1A* expression in GABAergic cells, they designed an AAV9 vector expressing compact engineered transcription factor (eTF*^SCN1A^*) under control of the modified *GAD1* promoter. More precisely, the upstream regulatory sequence used in this study contained the enhancer element (in the human genome, it is located ∼50 kb upstream of *GAD1*), the proximal promoter and the 5′-UTR sequence of *GAD1*, and a part of the first *GAD1* intron. To increase the cell-type specificity, a few miRNA binding sites that should prevent translation in excitatory neurons were added to the vector downstream of the transgene. The cassette containing all these regulatory elements and some other elements increasing transgene expression was named “RE^GABA^”. The specificity of this vector function in GABAergic neurons was evaluated in experiments with expression of the EGFP reporter gene, and the constitutive CBA promoter was used for comparison. The expression of EGFP under the control of RE^GABA^ was limited to cells that coexpressed the GABAergic neuronal markers—GAD67, PV, and SST—while the expression of EGFP under the control of the CBA promoter was detected throughout the brain in multiple cell types. The expression of eTF*^SCN1A^* under the control of RE^GABA^ increased *SCN1A* expression specifically within GABAergic neurons in vivo. SCN1A transcript levels in the cortical brain were ∼2-fold higher in GABAergic cells than in excitatory cells. GABAergic neurons containing the eTF*^SCN1A^* transcript expressed ∼30% more SCN1A than GABAergic cells without eTF*^SCN1A^*; this was sufficient for rescuing multiple disease phenotypes in a DS mouse model. Importantly, treatment with AAV9-RE^GABA^-eTF*^SCN1A^* did not elevate *SCN1A* expression in excitatory cells [14].

In recent years, an alternative approach for driving viral transgene expression specifically in GABAergic neurons was developed that uses specific enhancers rather than promoters [56]. It is described in Section 4.1. Another alternative is to combine a constitutive promoter with miRNA target sites that provide cell type specificity [150]. This is described in Section 4.2.

#### 2.2.5. Astrocyte-Specific Promoters: GFAP and Its Derivatives

The role of glial cells in the pathogenesis of epilepsy is often overlooked. However, recent studies show that astrocyte dysfunction contributes to epileptogenesis and the expression of the epileptic phenotype, and there is a need for developing viral vectors for gene therapy targeting astrocytes [27]. Another important consideration that makes astrocytes an attractive choice for gene therapy of epilepsy and some other diseases is their abundance (astrogliosis) when the number of neurons is depleted due to neurodegenerative processes [96].

Notably, not all viral vectors that are used for neurons are capable of astrocytic transduction. There are reports of LV, AAV5, AAV8 and AAV9 vectors being successfully used for gene delivery to glial cells [96]. Interestingly, AAV9, when injected i.v., preferentially targets neurons in neonatal mice, but it targets astrocytes in adult mice [151].

For targeted transgene expression in astrocytes, the *GFAP* gene promoter or its shortened versions may be applied. Glial fibrillary acidic protein (GFAP) is a cytoskeletal intermediate filament protein expressed in mature astrocytes and radial glia, while neurons utilize different mechanisms to suppress GFAP expression. Thus, GFAP is the most popular astrocyte marker. AAV vectors carrying different versions of the GFAP promoter are extensively used in research to target astrocytes in rodents and primates. The conventional human GFAP promoter (also called gfa2) is 2.2 kb long. The limited packaging capacity of AAV vectors called for the search of truncated versions of gfa2. By removing different enhancing regions, gfa2 modifications with the size of 1.74 kb (robust and highly selective gene expression in murine astrocytes in vivo) and even 448 bp (gfa28 promoter, with very restricted expression in the mouse brain and broad transduction potential for both astrocytes and neurons) were generated by different groups. It was later found that very short sequences at the beginning of the enhancer C within gfa2 may determine region- and cell-specific transgene expression. The insertion of the initial fragment of enhancer C into the shortest gfa28 promoter restored its characteristics to the level of the full-length gfa2. This novel 681 bp promoter, named gfaABC_1_D or gfa104, displayed greater activity but comparable expression patterns as the full-length gfa2 in transgenic animals. Multiple studies confirmed the efficiency of the gfaABC_1_D promoter for gene expression in astrocytes in vivo using AAV-based delivery (reviewed in [52]).

Young et al. tested the capacity of the GFAP promoter to direct and restrict transgene expression to astrocytes. They chose glutamine synthetase (GS) and excitatory amino-acid transporter 2 (EAAT2) as putative therapeutic transgenes. In a healthy brain, astrocytes mediate the clearance of glutamate diffusing out from the synaptic cleft via EAAT1 and EAAT2 glutamate transporters. Astrocytic glutamate is then converted to glutamine by GS before being shunted back to neurons. The disruption of glutamate clearance would lead to glutamate receptor overactivation and excitotoxicity. Another transgene tested in this study was miR-ADK, an miRNA targeting adenosine kinase (ADK). ADK is the astrocyte-based key negative regulator of the endogenous anticonvulsant adenosine; the overexpression of this kinase in the brain of Adk-tg mutant mice causes spontaneous epilepsy [27]. First, the GFAP promoter was used in AAV9 or AAV8 vectors expressing GFP reporter. The viruses were injected into the dorsal hippocampus of rats. Most GFP+ cells were identified as astrocytes based on their location, morphology and GFAP immunoreactivity. Some GFP+ cells were neurons (NeuN+), while no microglial (Iba1+) cells expressed the reporter. The total volumes of the hippocampus transduced by both vectors were similar, but GFP+ cells were more homogeneously distributed throughout the dorsal hippocampus in AAV9-injected rats, so this serotype was chosen for further experiments with the chosen transgenes. The overexpression of GS or EAAT2 did not lead to significant changes in symptoms of epilepsy in the intrahippocampal kainate infusion model. Moreover, the expression levels of both these proteins were not significantly increased, so apparently the GFAP promoter was too weak for the chosen application. However, miR-ADK expressed under the control of the same promoter caused decreased levels of ADK and a decreased duration of kainate-induced seizures [96].

Theofilas et al. also manipulated the expression level of ADK. They used Adk-tg epileptic mice. The authors developed AAV8-based vectors with the astrocyte-specific gfaABC_1_D promoter. The payload was cDNA of the cytoplasmic isoform of ADK in the antisense orientation, and the downregulation of the genomic *Adk* expression was achieved by RNA interference. The virus (Adk-AS) was injected unilaterally into the CA3 region of the hippocampus. The resulting decrease in ADK expression was modest (about 4%) but significant (compared to the contralateral side). Dual immunolabeling with antibodies to ADK and GFAP confirmed that ADK expression was preferentially knocked down in astrocytes. EEG data showed that Adk-tg mice treated with Adk-AS demonstrate a substantial decrease in seizure activity at the side ipsilateral to the virus injection site (only one of four mice in the treated group had seizures). In the same study, a similar construct but with *Adk*-cDNA in the sense orientation (Adk-SS) was used to overexpress ADK in the astrocytes of WT mice. It was demonstrated that this overexpression alone, in the absence of astrogliosis or any other epileptogenic event, is sufficient to trigger electrographic seizures. Therefore, the overexpression of ADK may be a cause of chronic recurrent seizure activity, and the *Adk* gene is a rational target for therapeutic intervention [27].

Zheng et al. used an AAV9 vector with the human GFAP promoter to express transcription factor NeuroD1 and induce astrocyte-to-neuron conversion. The virus was injected in rat hippocampus 1 week after pilocarpine-induced SE (a model of TLE). The gradual conversion of astrocytes into neurons over 9 weeks post-injection was confirmed by GFAP and NeuN immunostaining. NeuroD1-converted cells were also immunopositive for various GABAergic neuron markers—GAD67, PV, nNOS, SST, CR or CB (calbindin). Moreover, many of the infected cells were also stained positive for Lhx6, a transcription factor responsible for the generation of GABAergic neurons during brain development. Electrophysiological recordings confirmed that NeuroD1-converted neurons show typical features of interneurons and can integrate into functional neural circuits by establishing synaptic connections with other neurons. The viral expression of NeuroD1 reduced spontaneous seizures and rescued multiple behavioral abnormalities observed in rats with TLE. The authors also note that NeuroD1 expression is only needed transiently for astrocytes to convert into neurons; when they become mature neurons, NeuroD1 is no longer needed, which suggests that the viral expression does not have to be sustained long-term in this case [97].

The key properties of cell type-specific promoters relevant for gene therapy of epilepsy are listed in Table 4.

To summarize the chapter on cell type-specific promoters in the context of epilepsy, we must admit that the existing solutions do not provide perfect specificity, especially for glutamatergic neurons. The CaMKII promoter also functions in inhibitory neurons, while the VGLUT1 promoter is not active in a large fraction of glutamatergic neurons that express VGLUT2. Targeting interneurons may be even more complicated considering the diversity of their subtypes (but see Section 4.1). Promoters with limited specificity for principal neurons or interneurons may still be useful in specific cases of epilepsy, depending on the location of the epileptic focus. The use of astrocyte-specific promoters, and the targeting of astrocytes in the epilepsy treatment in general, are promising, but this direction of research is still new. The astrocyte-to-neuron conversion demonstrated in [97] is particularly interesting, since it proposes a way to regenerate functional GABAergic neurons in TLE patients without the need for an external cell bank.

### 2.3. Drug-Dependent Promoters

Gene therapy has an inherent problem relating to dose; the distribution of transfected cells and the expression level of the target gene cannot be changed after the therapy has been given, which could lead to under- or over-dosing [8]. Importantly, immune responses to AAV vectors may prevent re-dosing with the same therapy [116]. Among the ways to circumvent this problem are optogenetics (vectors expressing ion channels stimulated by light) and chemogenetics (vectors expressing artificial receptors stimulated by designer drugs). This way, the function level of the target protein may be manipulated by a second factor [8]. But a drug-regulated approach may be utilized not only to open and close modified receptors, but also to regulate the expression level of any protein, if the drug affects the promoter in the DNA vector. Conveniently, promoters regulated by exogenous molecules are already widely used in developing genetically modified mouse strains that allow researchers to turn the expression of a gene driven by such a promoter on and off when it is required. The most commonly used drug-dependent promoters are regulated by tetracycline and its derivatives, like doxycycline [120,156], but there are also promoters regulated by streptogramin, coumermycin and novobiocin, ecdysone, rapamycin-related derivatives and other molecules [157]. A common property of antiepileptic drugs is the problem of a narrow therapeutic window between seizure control and side effects. Therefore, a long-term goal is to develop drug-inducible gene therapies that can be regulated by other drugs with less side effects [98].

#### 2.3.1. Tetracycline-Responsive Element

In *E. coli*, the natural Tet repressor protein (TetR) is constantly expressed and, in the absence of tetracycline, blocks the transcription of the genes of the tetracycline-resistance operon by binding to the tet operator (tetO). If tetracycline is present, it binds to TetR, which changes its conformation and is released from tetO. Bacterial TetR and tetO provide the basis of engineered gene expression regulation in mammalian experimental systems in vitro and in vivo. Tetracycline responsive element (TRE), a derivative of tetO, may be added upstream of a minimal universal promoter. TetR was modified to make an artificial transcriptional activator (tTA) that should be constitutively expressed from a separate promoter. In the Tet-Off system, tTA is constantly bound to TRE and activates gene expression when the drug is absent. In the Tet-On system, the artificial regulatory protein is a mutated “reverse” Tet activator (rtTA) that can only bind TRE and activate the gene expression when the drug is added [120,156].

Haberman et al. designed AAV vectors with a doxycycline-off element expressing the galanin coding sequence with or without FIB (AAV-FIB-GAL or AAV-GAL). In vitro, only cells transfected with AAV-FIB-GAL produced substantial amounts of galanin. AAV-FIB-GAL was then injected into rat inferior colliculus, and after a few days the seizures were triggered by electrical stimulation of the same area. Over a 4-week testing period, the stimulation threshold for seizure genesis increased significantly in the AAV-FIB-GAL-treated group. When doxycycline was added to the drinking water, the threshold for seizure genesis returned to baseline levels within 1 week. Upon the removal of the doxycycline, the seizure-suppressive effects slowly increased over a few weeks [21]. Interestingly, when the same research group tried a similar approach with a virus expressing the antisense *NMDAR1* sequence, tetracycline-dependent promoter was ineffective in treating seizures, and even had the opposite effects, while the CMV promoter worked as expected. Patterns of expression driven by CMV and Tet-Off promoters overlapped only partially, which means that both promoters have some degree of cell-type specificity [68] (also described in Section 2.1.1).

In a recent study by Sullivan et al., an AAV vector with a doxycycline-regulated element was tested. They used the SYN promoter together with the Dox-On genetic switch activated by doxycycline. This virus expressed an engineered version of the leak potassium channel *Kcnk2* (TREK-M) that inhibits neuronal firing, and it was injected into rat hippocampi. The rats were then subjected to the electrical kindling of seizures. Treating the epileptic rats with doxycycline successfully reduced spontaneous seizures. However, even the low expression of TREK-M in the absence of doxycycline was sufficient to cause rats to develop spontaneous recurring seizures. Localization studies of infected neurons suggest that seizures were caused by TREK-M expression in GABAergic inhibitory neurons. In contrast, doxycycline increased the expression of TREK-M in excitatory neurons, thereby reducing seizures through the net inhibition of firing. This study highlights the challenges of controlling off-target expression [98].

In some cases, the Tet-Off system is simply used for constitutive gene expression, and tetracycline is not added. Thompson et al. [99] engineered immortalized mouse cortical neurons and glia to make them produce GABA. To achieve this, the cells were transfected by a LINX vector (based on encephalomyocarditis virus [158]) expressing GAD65 (a variant of glutamate decarboxylase) under the control of the Tet-Off element. Transfected cell lines showed GAD65 mRNA expression, enzymatic activity, and GABA release. GABA-producing cells were then transplanted into the rat brain, into either anterior or posterior substantia nigra (SNr). After recovery, the rats were subjected to electrical kindling. GABA-producing cells affected the kindling rate, depending on the implantation region. The transplantation of GAD65-expressing cells into the posterior SNr significantly reduced the number of stimulations needed to produce kindling, thus increasing seizure susceptibility. By contrast, the transplantation of these cells into the anterior SNr increased the number of trials needed to produce kindling (though not significantly when compared to control animals receiving β-gal-producing cells) [99].

#### 2.3.2. Drug-Dependent Protein Expression Control: An Alternative to TRE on the Posttranscriptional Level

The Tet-On expression system relies on the expression of a foreign expression regulator protein that can trigger immune responses in nonhuman primates. As a result, currently, there are no approved clinical applications based on this system [159,160]. This is why it is necessary to continue the search for alternative drug-dependent regulators that may effectively tune gene expression to the desired level [161]. In the review by Wang et al. [162], different approaches to the transcriptional, posttranscriptional and posttranslational regulation of AAV gene expression are described in detail. Here, we will provide an example of one promising method.

Very recently, Luo et al. [160] designed a novel RNA-based tetracycline-induced expression regulator they called the polyadenylation (pA) switch. In this system, PAS is added into the 5′-UTR of the transgene, which (in the absence of tetracycline) causes the cleavage of the transcript and prevents the expression of the transgene. The PAS is embedded within an aptamer—a secondary RNA structure within the 5′-UTR. The aptamer is shaped so it can bind tetracycline. Tetracycline bound to the aptamer clamps the PAS and prevents it from being recognized by the cleavage and polyadenylation machinery, so the downstream sequence can then be expressed. The authors also added the second regulation mechanism—controlling PAS by an inducible alternative splicing. The 5′-UTR sequence is designed in such a way that the binding of tetracycline by the aptamer causes the bypass of the nearby 3′-splicing site and the usage of a downstream alternative site. This leads to the complete removal of the PAS-containing fragment from the pre-mRNA, therefore inducing gene expression. Adding this mechanism increased the sensitivity of the pA switch to tetracycline and dramatically lowered the EC_50_ of tetracycline to 0.5 μg/mL (this is within the FDA-approved dose range, which is very important for potential clinical applications). The authors confirmed the efficiency of the pA switch in multiple experiments in vitro using expression plasmids or genome integration, with different promoters and reporter genes in different cell lines. For the experiment in vivo, the pA switch was added to the AAV2/9 vector with the CMV promoter and luciferase reporter. The virus was injected into the hind leg muscles of mice. Mice exhibited low leakage baseline expression before the tetracycline treatment, and tetracycline caused a dose-dependent induction of luciferase expression (up to 300-fold with 80 mg/kg tetracycline). Approximately 3 months later, it was confirmed that the luciferase expression had returned to basal levels, and a second dose of tetracycline caused similar expression induction. Thus, the effect of tetracycline on the switch is reversible and repeatable. The main advantage of the pA switch is that it achieves expression induction levels similar to those of systems that use foreign proteins (like tTA), but with no risk of immune response caused by such proteins. Another advantage is that this switch requires no specialized promoter elements and may be coupled with various promoters for tissue-specific applications [160].

Using drug-dependent promoters for gene therapy of epilepsy would allow one to finely regulate the expression of the desired transgene; however, it seems like the most commonly used drug-dependent regulatory element, TRE, is not suitable for the clinical application because the Tet-On system is regulated by an artificial protein, which may be immunogenic. The Tet-Off system has the same problem, and would be even less suitable for patients because it requires the daily intake of an antibiotic when the transgene expression is not needed. However, the RNA switch proposed in [160] circumvents the need for tTA/rtTA expression, and also requires lower doses of tetracycline; this solution is very recent, and it requires further investigation.

### 2.4. Activity-Dependent Promoters

Activity-dependent promoters, typically of immediate early genes (IEGs), respond rapidly to increases in neuronal activity. In neuroscience, the term “IEG” is applied to genes that reach the peak level of their expression in 30–60 min after a relevant stimulation [163]. Thus, these genes may be used to identify hyperactive cells during seizures, and it seems promising to use their promoters to drive a therapeutic transgene expression.

#### 2.4.1. cfos Promoter

The *Fos* (previously called *c-fos*) gene encodes the c-Fos protein (Fos proto-oncogene), a part of the activator protein-1 (AP-1) transcription activation complex. c-Fos expression is a marker of cellular activity (for both neurons and glia) and may be stimulated by a wide range of stimuli. In neurons, the induction of c-Fos expression is involved in both normal function (reaction to excitatory synaptic transmission, long-term potentiation) and response to stress on the cellular level. Importantly, seizure induction led to a massive increase in c-Fos expression in different models [163].

Qiu et al. used the cfos promoter in an AAV9 vector expressing EKC to reduce neuronal excitability in a closed-loop system. Experiments in vitro and in vivo with a similar construct expressing dsGFP reporter confirmed that the reporter expression follows network dynamics. Cultures transfected with *cfos*-EKC showed less spontaneous network activity compared with cultures transfected with *cfos*-dsGFP, confirming the ability of EKC to decrease neuronal and network activity. In experiments in vivo, *cfos*-EKC was injected into the hippocampus. Two weeks later, a single generalized seizure was induced by pentylenetetrazole (PTZ). Mice were sacrificed after 2 h to prepare acute hippocampal slices for electrophysiological recordings. Neurons transfected with *cfos*-EKC exhibited a profound decrease in excitability compared with *cfos*-dsGFP-treated neurons. To test the effect of *cfos*-EKC on the outcome of repeated chemoconvulsant challenge, the authors expressed *cfos*-EKC in murine hippocampi and performed three consecutive PTZ injections. The first injection triggered a generalized seizure with no difference between the *cfos*-EKC group and the control group (injected with *cfos*-dsGFP). This was expected because the *cfos* promoter is inactive under baseline conditions. The second PTZ injection 24 h later led to markedly attenuated seizures in the animals injected with *cfos*-EKC. After the third PTZ injection 2 weeks later, the difference between the groups disappeared again. It was also expected because, by this time, the overexpression of EKC was predicted to return to baseline. Thus, the activity-dependent induction of EKC expression protects against repeated seizures, but in a time-limited manner. More promising results were obtained in experiments with a clinically relevant model of chronic epilepsy. KA was used to induce a period of SE; after 2 weeks, mice started to exhibit spontaneous seizures. After the viral expression in murine hippocampi, *cfos*-EKC-treated animals showed a robust decrease in the number of seizures compared with mice receiving *cfos*-dsGFP. Thus, in the chronic model, the antiepileptic effect of *cfos*-EKC treatment was persistent. Immunochemistry data show that interictal discharges in KA-treated mice are sufficient to repeatedly reactivate the *cfos* promoter and support the expression of the reporter even in the absence of overt seizures. Interestingly, promoters of other IEGs and another potassium channel gene (*KCNJ2*) tested in the same study were less effective in decreasing excitability [100].

#### 2.4.2. Arc Promoter Elements as a Part of a Hybrid Promoter

Immediate early gene *Arc* (activity-regulated cytoskeleton-associated) encodes a protein that is crucial for long-term plasticity in neurons. Arc plays a critical role in AMPA receptor trafficking and has been implicated in actin polymerization at synapses [164]. Arc mRNA expression was shown to be induced as early as 5 min after seizures caused by electric shock if the current was high enough [165].

Burke et al. designed an artificial promoter they called the “epilepsy promoter”, or EpiPro. The core EpiPro promoter consists of multiple copies of response elements that bind extracellular and intracellular factors upregulated in epilepsy—the transcriptional complex consisting of nuclear factor of activated T cells and AP-1 (NFAT-AP1), early growth response protein (EGR), Ca^2+^, and cyclic adenosine monophosphate (cAMP). Each of these response elements alone can drive the expression of reporter genes. In the construction used in this study, EpiPro is flanked with “Arc elements” (distal and proximal synaptic activity-responsive elements from the *Arc* promoter). All these elements were combined to target the majority of epileptic neurons that may represent different cell types. Experiments with mammalian cell lines confirmed that core EpiPro is regulated by cAMP and CREB. Adding distal Arc elements increased the reporter expression approximately 5-fold. The authors then tested the efficiency of the EpiPro-containing, GFP-expressing AAV vector (scADGFP) in rodent models of epilepsy. The virus was injected into different parts of the hippocampus. In the rat lithium/pilocarpine chronic epilepsy model, SE caused a 10-fold increase in both the number of GFP+ neurons and the intensity of GFP fluorescence compared with the control animals (who were injected with the same virus but not treated with Li/pilocarpine). The most striking difference in GFP expression was observed in the granule cell layer of the dentate gyrus. To test if these GFP+ neurons are responsible for the seizures, the authors cotransfected epileptic animals with scADGFP and an AAV expressing TREK-M, a mutant K^+^ leak channel known to reduce firing in cultured neurons and the severity of seizures in in vivo models of epilepsy. TREK-M inhibited GFP expression from scADGFP by 80%. To determine the time course of expression of scADGFP, the authors used the acute seizure mouse model with KA. Similar to the results in chronic epileptic rats, the number of GFP+ dentate gyrus granule cells was substantially higher after a KA-induced seizure. Measuring expression at different time points after the seizure showed that EpiPro is activated rapidly by seizures, and then GFP expression decays slowly, reaching baseline values in 14 days. In the mouse model of chronic epilepsy (electrical kindling of VGAT-Cre mice), it was shown that EpiPro-driven GFP expression diminishes after seizure with different rates in different brain regions. Notably, in dentate gyrus granule cells, fluorescence decay was much slower in axons than in cell bodies, and CA3 neurons did not show a significant decay in GFP expression even within 30 days after the last seizure. The authors conclude that EpiPro should be useful in driving anti-seizure gene therapies, providing basal expression long after a seizure and increased expression that lasts for weeks after a seizure. However, EpiPro has basal expression in hippocampal projection neurons without seizures, which would be a problem for its application. It would be necessary to avoid silencing these neurons completely, which could lead to unwanted cognitive side effects [101].

#### 2.4.3. GABRA4 Promoter

Another promoter that becomes active after SE belongs to the *GABRA4* gene that encodes the α4 subunit of the GABA_A_ receptor (GABRα4). Previous studies showed that in humans with TLE and in rodent models of TLE, the expression of the GABRα4 subunit is increased, while the expression of the GABRα1 subunit is reduced in the dentate gyrus. These changes begin within 24 h of pilocarpine-induced SE in rats and persist for months as these animals become epileptic. These subunit alterations are associated with marked changes in receptor function and pharmacology. Raol et al. aimed to restore the disrupted balance by overexpressing the rat GABRα1 subunit under the minimal (500 bp) human *GABRA4* gene promoter in an AAV2 vector. Injection of the virus into the dental gyrus of rats decreased the risk of developing spontaneous seizures after SE induced by pilocarpine. Surprisingly, in this study, GABRα1 subunit overexpression was observed in dentate gyrus for only the first 2 weeks after SE. The reduction in exogenous GABRα1 expression at 4 weeks suggests that the GABRA4 promoter in AAV vector is either silenced or transcriptionally downregulated [102].

Overall, activity-dependent promoters on the first glance seem to be very fitting for use in developing gene therapies of epilepsy because of their adaptive response to seizures. However, we could find very few reports about such promoters, and they all have their downsides. The cfos promoter requires a seizure or at least an interictal discharge to occur to be induced, so it cannot fully prevent the epileptiform activity. EpiPro, on the other hand, shows some basal expression even without seizures (as well as some undesired expression in inhibitory neurons, as seen in supplementary images in [101]). The GABRA4 promoter was silenced in a few weeks after the infection. Thus, apparently, genes that change their expression not immediately after the seizure may be more useful as a source of promoters for expression vectors aimed at compensating for pathological changes in epilepsy.

To create a system with negative feedback that prevents seizures, other methods may be used. For example, Hristova et al. [80] used the optogenetics approach with an implanted device emitting light when excessive neuronal activity is detected, while Lieb et al. [93] chose a glutamate-gated Cl^−^ channel from *Caenorhabditis elegans* as a transgene to provide neuronal inhibition in response to elevations in extracellular glutamate.

### 2.5. RNA Polymerase III Promoters for Expression of Small RNAs

A specific type of transgenes used for developing gene therapy of epilepsy is small RNAs. Short hairpin RNAs (shRNAs) or microRNAs (miRNAs) may be used to downregulate the expression of an endogenous target mRNA. Short guide RNAs (sgRNAs) may be used for precise genome editing using the CRISPR/Cas9 system, or its variations, such as CRISPRa.

Even though there are examples of the successful expression of small RNAs under RNA polymerase II promoters [96,166], the use of such promoters (pol II normally transcribes mRNAs) is not optimal for expressing short RNAs from artificial vectors, since the presence of significant single-stranded extensions can affect the efficacy of short RNAs. RNA polymerase III transcribes a variety of small RNA species, and pol III promoters have well-defined initiation and termination sites. Of these, type III promoters like the U6 snRNA promoter and the H1 RNA promoter function entirely outside the transcribed sequence (no part of the promoter is transcribed). U6 and H1 promoters are commonly used for producing short RNAs from expression vectors [167]. The H1 promoter favors adenine at the first position of the transcript, while the U6 promoter favors guanine [168]. Human U6 promoter (found in multiple human genes encoding U6 small nuclear RNA) is 244 bp long and contains three sequences required for efficient expression—TATA box, proximal sequence element (PSE) and distal sequence element (DSE). PSE and DSE are homologous and interchangeable with elements of the U2 snRNA gene, which is transcribed by pol II. Interestingly, crippling the TATA box allows U6 genes to be transcribed by pol II [169].

In gene therapy, it is possible to make sgRNA-dependent modifications cell type-specific by combining U6-driven sgRNA with a Cas9-like protein driven by a cell type-specific promoter, in the same or a different virus.

The *SCN8A* gene encodes the Na_v_1.6 subunit of voltage-gated sodium channels. It is broadly expressed throughout the nervous system and strongly modulates neuronal excitability. In rodents, *Scn8a* expression is increased in the hippocampus following SE and amygdala kindling. Thus, *Scn8a* is a promising therapeutic target for epilepsy. Wong et al. used shRNA to knock down *Scn8a* expression in the murine hippocampus. They developed an AAV10 construct containing shRNA-Scn8a downstream of U6 promoter. The same vector also contained EGFP reporter (driven by CMV promoter). Viruses were injected into the hippocampus 24 hours after KA administration. Of the shRNA-Scn8a treated mice, 9 of 10 were seizure-free during the 8-week EEG recording period, whereas all control animals treated with KA exhibited spontaneous seizures. Behavioral tests revealed that *Scn8a* knockdown reduces KA-induced hyperactivity. Histological analysis showed that *Scn8a* knockdown in the hippocampus did not prevent neuronal loss associated with epilepsy, but reduced reactive gliosis [103].

Aimiuwu et al. used an scAAV9 vector with U6 promoter to express an miRNA targeting an abnormal allele of the gene encoding dynamin-1 (*DNM1*) in mice. Patients with de novo variants of this gene suffer from DEE: they have therapy-resistant seizures, intellectual disability, muscular hypotonia and other neurological symptoms. Mice with homozygous “fitful” mutation of this gene (*Dnm1*^Ftfl/Ftfl^) show a similar phenotype with severe ataxia, developmental delay, and fully penetrant lethal seizures by the end of the third postnatal week. *Dnm1*^Ftfl/Ftfl^ or WT mice received scAAV9-*miDnm1a* or control (scAAV9-EGFP or saline) via i.c.v. injection at the day they were born (P0). Mice were observed for survival, seizure activity, and weight until P30. None of control-injected *Dnm1*^Ftfl/Ftfl^ mice survived past P19, while 75% of scAAV9-*miDnm1a*-treated mutant mice survived at least until P30. There was also a significant growth improvement in scAAV9-*miDnm1a*-treated mutant mice. Seizures and seizure-like behaviors (wild runs, Straub tail, prolonged vertical jumps with subsequent facial grooming, continuous jerking of the forelimbs and hindlimbs) were monitored for approximately 5 min every second day. While scAAV9-*miDnm1a*-treated and control *Dnm1*^Ftfl/Ftfl^ mice exhibited seizure behaviors, treated mutant mice had fewer overall observed events between P14 and P18. scAAV9-*miDnm1a* treatment also mostly improved the developmental outcomes of *Dnm1*^Ftfl/Ftfl^ mice assessed by measuring locomotion, grip strength, negative geotaxis and similar parameters. On the histological level, it was confirmed that scAAV9-*miDnm1a* treatment rescued fibrillary gliosis (assessed with anti-GFAP antibodies), neurodegeneration (assessed with Fluoro-Jade C staining) and abnormal neuronal activity (assessed with anti-NPY and anti-c-Fos antibodies) observed in the hippocampus of *Dnm1*^Ftfl/Ftfl^ mice at P18. By P30, most histological parameters of scAAV9-*miDnm1a*-treated *Dnm1*^Ftfl/Ftfl^ mice were comparable to those of WT controls [104]. More recently, the same research group improved this strategy—they combined the knockdown of a pathogenic *Dnm1* allele with the ectopic expression of normal *Dnm1*. As a model of DEE, they expressed a patient-based mutant variant of *Dnm1* in GABAergic neurons of mice, which resulted in growth delay and lethal seizures evident by postnatal week three. Then, an AAV9 vector expressing both miRNA targeting the mutant allele (under U6 promoter) and the normal *Dnm1* cDNA (under SYN promoter) was designed. When injected i.c.v. to newborn pups, the bivalent virus improved both survival and growth of the mutant mice. Interestingly, viruses expressing either the miRNA or the cDNA alone did not rescue the mutant phenotype [105].

In some cases, the CRISPR method may prove useful for gene therapy, when a deletion of a gene has a therapeutic effect. Interestingly, this effect may be achieved by the cell type-specific deletion of a normal (not disease-associated) allele. Guan et al. utilized CRISPR to delete the *Alox5* gene in murine hippocampal neurons and assessed the effects of this in two models of epilepsy. *Alox5* encodes 5-lipoxygenase, a key enzyme that produces leukotrienes, pro-inflammatory mediators. Previous research has shown that 5-lipoxygenase is involved in post-seizure brain injury. The AAV9 virus designed for this study expressed SaCas9 under the SYN promoter and sgRNA targeting *Alox5* under the U6 promoter (AAV9-hSyn-SaCas9-U6-sg*Alox5*). A separate virus expressing the EGFP reporter under the SYN promoter was mixed with it. The control group received a similar virus mix, but with a different sgRNA sequence. In one set of experiments, viruses were injected into the mouse hippocampus 4 weeks before the induction of seizures with either pilocarpine or KA. In both models, AAV9-hSyn-SaCas9-U6-sg*Alox5* increased the first seizure latency (which indicates the decrease in seizure susceptibility) and decreased different parameters of seizure severity. In the pilocarpine model, EEG recording for 3 consecutive days also showed a decrease in some seizure-associated parameters in AAV9-hSyn-SaCas9-U6-sg*Alox5*-treated animals. In another set of experiments, viruses were injected a few hours after the pilocarpine administration, and various analyses started 42 days after the injections. Continuous video and EEG recording for a week showed that AAV9-hSyn-SaCas9-U6-sg*Alox5* decreased the number of spontaneous recurrent seizures and improved some EEG parameters associated with epilepsy. Histological analysis showed that the virus improved molecular pathologies involved in epileptogenesis—neuronal loss, neurodegeneration, astrogliosis and mossy fiber sprouting. Behavioral tests showed that the neuron-specific deletion of *Alox5* improves diverse neuropsychiatric comorbidities observed in this model of epilepsy, especially anxiety, cognitive deficit and autistic-like behavior (assessed by social interaction test). Additional experiments that aimed to unravel the mechanisms of anti-epileptic effects of *Alox5* deletion showed that, unexpectedly, it is not linked to inflammation. Instead, it was found that Cas9-mediated *Alox5* knockout resulted in a decrease in glutamate level in the pilocarpine epilepsy model [106].

CRISPR activation (CRISPRa) is a variant of CRISPR that provides a possibility to directly target promoters and modify the expression of endogenous genes. The CRISPRa system consists of a nuclease-defective Cas9 fused to a transcription activator (dCas9a) and an sgRNA that targets dCas9a to the promoter of the gene of interest. Colasante et al. utilized this method to increase the expression of *Kcna1*. In this work, many different LV and AAV constructs were used for different experiments, including combinations of two AAVs expressing sgRNA and dCas9a (these viruses also expressed reporter genes). In all vectors, sgRNA expression was driven by the U6 promoter, while for other transgenes, EF1α, SYN, CaMKII, or tetracycline-dependent promoters were used. It was confirmed that the CRISPRa system may be successfully applied to upregulate *Kcna1* expression and decrease neuronal excitability in vitro and in vivo. Moreover, the endogenous *Kcna1* overexpression achieved by this method alleviated symptoms in the intra-amygdala KA model of chronic epilepsy in mice. Epileptic animals injected with Kcna1-dCas9A viruses intrahippocampally showed decreased seizure frequency and had less cognitive deficits. Additionally, the RNA-Seq analysis of hippocampi showed that Kcna1-dCas9A was able to rescue transcriptomic changes in this epilepsy model as well [28].

Another study used the CRISPRa method to manipulate expression levels of K-Cl cotransporter isoform 2 (KCC2). KCC2 is a critical modulator of neuronal excitability, and KCC2 expression is usually dysregulated in epilepsy. The authors used two AAV2/9 vectors expressing halves of dSaCas9 fused with transcriptional activators. Two halves of the fusion protein were expressed under the CaMKII promoter in both viruses; the virus with the N-part of dSaCas9 also contained the sgRNA under the U6 promoter. The split-intein system allowed two halves of the fused protein to be assembled post-translationally. The viruses were injected into the subiculum of adult mice (the subiculum was shown to be involved in drug-resistant seizures). In this study, various models of epilepsy were used—rapid hippocampal kindling (the viral treatment performed after mice acquired stable generalized seizures), acute KA-induced SE (KA administered in the hippocampus 2 weeks after the viruses) and chronic spontaneous seizures (KA administered 2 months before the viruses). Western blots confirmed that the CRISPRa system used in this work effectively upregulated KCC2 expression in the subiculum. In the hippocampal kindling model, EEG showed that the upregulation of KCC2 expression by the viruses attenuated the severity of generalized seizures. Moreover, the viral treatment increased the sensitivity to low doses of diazepam (as compared to anti-seizure diazepam effects in the same animals before virus injection). In the KA-induced SE model, diazepam is usually ineffective if administered ∼30 min after SE induction. Experimental animals received diazepam 30 or 60 min after SE, and EEG was recorded for 180 min after this. It was shown that in the group treated by the viruses, the average time of SE termination was shorter, and the percentage of SE-free animals was higher. The results prove that the upregulation of KCC2 expression prevented diazepam resistance in refractory SE. In the KA-induced chronic epilepsy model, sensitivity to valproate was tested. The EEG of freely moving mice before the virus injection confirmed that valproate had little effect in this model. After the viral treatment, the same valproate dose significantly decreased seizure frequency and duration in the same animals. In an additional experiment, the authors used calcium fiber photometry to estimate the effects of CRISPRa-induced KCC2 overexpression on GABAergic neuron function. To do this, a combination of *Vgat-cre* mutant mice, AAV-CaMKIIα-GCaMP6s (expressing Ca^2+^ reporter) and AAV-hSyn-DIO-ChrimsonR-mCherry (expressing photosensitive ion channel) was used. This was necessary to simultaneously activate GABAergic neurons and record the activity of “CaMKIIα neurons” (neurons expressing the transgene from the CaMKII promoter, presumed to be excitatory). In non-epileptic animals, CaMKIIα neurons were prominently inhibited by GABAergic neurons. Once the mice were kindled, this was no longer observed, suggesting that the inhibitory function of GABAergic transmission has degraded. However, if these animals were additionally injected with the viruses upregulating KCC2 expression, CaMKIIα neurons were partly inhibited when GABAergic neurons were activated, indicating that upregulating KCC2 could partly reverse the recurrent seizure-induced impaired GABA neurotransmission [92].

Yamagata et al. used a similar method to rescue the phenotype of mice with *Scn1a*-haplodeficiency (genetic model of DS). Since aggravating effects of increased *Scn1a* expression in excitatory neurons were suggested, the authors used a complex method to restrict the expression of dCas9-VPR to inhibitory neurons. The CRISPR-ON system used in this paper consisted of *Vgat*-Cre (Cre recombinase expressed under Vgat promoter specifically in inhibitory neurons), Cre-controlled dCas9-VPR (nuclease-defective Cas9 fused with multiple transcriptional activators), and sgRNAs targeting the *Scn1a* promoter. dCas9-VPR and *Vgat*-Cre were inserted into the mouse genome (floxed-dCas9-VPR^VPR/+^/*Vgat*-Cre^Cre/+^/*Scn1a*^RX/+^ triple mutant mice were generated). Multiple *Scn1a* promoter-specific sgRNAs were designed and tested in preliminary experiments with cell lines. As a result, the four most effective sgRNAs were selected; all of them target the upstream promoter region of *Scn1a* (which has two alternative TSSs corresponding to two promoters) and were shown to have synergistic activity. An AAV vector expressing the combination of these four sgRNAs was designed wherein each sgRNA had its own U6 promoter (AAV-4xU6-sgRNA). The virus was injected i.v. to 4-week-old mice. Immunohistochemistry at 14 weeks showed that increased Na_v_1.1 immunoreactivity was mainly observed in the PV+ neurons in the neocortex, hippocampus, reticular thalamic nucleus, and cerebellum of the animals treated with AAV-4xU6-sgRNA. Importantly, the age of 4 weeks is later than the typical onset of epileptic seizures in *Scn1a*^RX/+^ mice, and most of the triple mutants actually died before the virus injection. Moreover, triple mutants showed much lower survival rates than the group that only had the *Scn1a*^RX/+^ mutation, and the floxed-dCas9-VPR^VPR/+^/*Vgat*-Cre^Cre/+^ mice also showed partial lethality, suggesting that the dCas9-VPR itself is toxic. All the triple mutant mice that received AAV-4xU6-sgRNA survived to the end of the observation period (while additional deaths were observed in the *Scn1a*^RX/+^ and flox-dCas9-VPR/*Scn1a*^RX/+^ groups at the same age). Additionally, AAV-4xU6-sgRNA treatment partially ameliorated heat-induced seizures in these mice and reduced spontaneous spike discharges (as measured by ECoG observation in freely moving animals for 3 days). Behavioral tests showed that the AAV-4xU6-sgRNA treatment of triple mutants partially improved behavioral abnormalities associated with the *Scn1a*^RX/+^ genotype—hyperactivity, decreased anxiety and loss of preference for social novelty. Taken together, these data suggest that increased *Scn1a* expression in inhibitory neurons of *Scn1a*-haplodeficient mouse brain achieved by dCas9-VPR prevents sudden death, partially alleviates seizures and improves associated behavioral symptoms. However, the results of this study cannot be easily translated to clinical application because a part of the gene therapy machinery was incorporated into the mouse genome. Moreover, components of this machinery proved to be toxic for mice. And finally, most of the triple mutants died before receiving the treatment, and it could be argued that mice in this group who survived longer are intrinsically different from the ones who died early [29].

The U6 promoter was also used in the study by Qiu et al., described in Section 2.4.1. These authors used the CRISPRa system for an additional experiment to determine whether cfos-driven constructs may dampen the effect of a proconvulsant manipulation in vitro. The lower baseline activity in *cfos*-EKC-treated cultures precluded a simple comparison with *cfos*-dsGFP. To have similar baseline activity levels between experimental groups, CRISPRa was used in activating the *Kcna1* gene promoter. The components of the system were delivered in two AAV9 vectors. One virus carried dCas9a driven by Tet-On promoter. The other virus expressed rtTa and EGFP, both driven by cfos promoter, and U6-driven sgRNA targeting the endogenous *Kcna1* gene. The control vector carried sgRNA targeting the yeast *LacZ* gene instead. In this system, endogenous *Kcna1* overexpression (which should decrease neuronal excitability) happened only when (1) neurons were co-transduced with both AAV9s, (2) doxycycline was applied, and (3) activity was sufficient to activate the cfos promoter. In these conditions, the baseline network activity between control and experimental cultures was similar, but picrotoxin application revealed a clear difference—picrotoxin failed to increase network activity in cultures treated with *cfos*-CRISPRa_*Kcna1*, confirming the ability of cfos promoter-driven vectors to follow neuronal dynamics and decrease network hyperexcitability in vitro [100].

As a conclusion, small RNAs may be very useful for gene therapy because of their high specificity and small size, so the same vector may carry a combination of small RNAs tailored to a specific cell type or pathological state. shRNAs or miRNAs may be successfully used for gene therapy in cases when it is necessary to prevent the endogenous overexpression of some gene. However, there are examples wherein the expression of a full-size antisense transcript may also work for this purpose [27,68]. CRISPRa seems to be an elegant prospective method to stimulate the expression of endogenous alleles without editing the host genome or introducing foreign therapeutic transgenes. But importantly, this method requires the expression of dCas9 fused with a transcriptional activator; this protein itself is foreign and may even be toxic, as was observed in [29], so it is possible that adverse side effects would outweigh the benefits of using this system.

## 3. Time Course of Transgene Expression Driven by Different Promoters; Promoter Silencing

Ectopic gene expression by replication-defective viral vectors is supposed to be transient, meaning that the vector would eventually degrade. The time of viral expression in experiments depends on the cell turnover rate in the tissue, with new cells “diluting” transfected cells since episomes are usually lost during mitosis. There is also a possibility of an immune response triggered by any component of the virus, causing faster expression inactivation [116]. Even if the virus is intact, the transfected cell may silence its expression. There is a tendency for some promoters to turn off in cells in which they are not normally active [54,116,170]. For example, in the viral vectors used for the CNS, the CMV promoter initially delivers high levels of neural expression but is silenced by 10 weeks in the murine spinal cord and dorsal root ganglia [54]. Even in experiments wherein transgenes are supposed to be integrated in the cell genome, their expression is often silenced by diverse mechanisms [171,172]. Activity-dependent promoters that are particularly interesting for treating epilepsy are a special case; they require “reactivation” and do not drive gene expression without it [100]. When designing vectors for gene therapy, it is very important to consider the estimated duration of the target gene expression.

In experiments in vivo, AAV vectors are known to maintain stable transgene expression in different tissues for months [173]. There are some reports that AAV expression in vivo may be conserved for years [116,117,174]. Notably, there is evidence that such long-term expression may be due to some viruses eventually becoming integrated into the cell genome, an undesirable effect of gene therapy linked to the risk of malignant transformation [116,170,175] (for AAV, integration frequency was estimated using ITR-seq as 1 event per 100 cells in primate hepatocytes) [170].

Unfortunately, in many works describing novel promoters proposed for gene therapy of neuropathologies, the time of observation is limited to a few weeks [54,61,100,101]. On the other hand, if a specific vector is already tested in vivo in some model of neuropathology, the therapeutic effect and the transgene expression may be monitored for months or even years [36,117,176,177,178]. Here, we will describe some of the papers on neuron-specific promoters wherein the time course of the transgene expression was measured for longer periods.

Hollidge et al. studied the time course of the vector DNA degradation and reporter (GFP) transcription and translation after the intrastriatal injection of AAV9 viruses in mice. The authors compared a ubiquitous CAG promoter with two neuron-specific promoters, SYN and CaMKII. The observation time was limited by 6 months. It was established that vector DNA rapidly decreases 10-fold over the first 3 weeks following injection, as it assembles into stable circular episomes and concatemers. This process was continually dynamic up to 3 months after injection. GFP protein expression with the CAG promoter was highest at 3 weeks, and then decreased, with similar levels at 3 and 6 months. Surprisingly, GFP mRNA levels continued to increase from 3 weeks to 3 months, despite the GFP protein expression decreasing during this time. GFP protein expression with the SYN promoter increased more slowly, reaching a maximum at 3 months. Importantly, transgene expression driven by the SYN promoter was continuing to rise even at the final 6-month time point. CaMKII promoter provided neuron-specific GFP expression that was weak over the entire 6 months [126]. The activity of the CAG promoter in AAV for at least 6 months was also shown in rats in a preclinical study of an epilepsy gene therapy vector. In this work, an AAV expressing the combination of NPY and its receptor Y2 under the CAG promoter was administered to healthy rats and showed no long-term adverse effects [176].

Klein et al. monitored GFP transgene expression levels in rat hippocampus and substantia nigra for 25 months (time interval comparable with rat life expectancy) after injections of an AAV2 vector with NSE promoter in these regions. GFP expression levels at 25 months were similar to those previously observed between 1 and 12 months using the same promoter and vector. GFP did not colocalize in GFAP+ astrocytes at any time interval [123].

Husain et al. compared three different promoters in AAV1 vectors expressing β-glucuronidase (GUSB): hGUSB promoter (ubiquitious), NSE promoter and latency-associated transcript (LAT) promoter from herpes simplex virus (which drives gene expression more efficiently in neuronal cell lines). The vectors were injected into the mouse cortex, striatum, hippocampus and thalamus, and GUSB expression in the brain was estimated at the 2- and 6-month time points. At 2 months, all three vectors showed mostly similar patterns of spatial distribution and similar overall amounts of vector mRNA. At 6 months after injection, GUSB enzymatic activity on the ipsilateral side was similar for all 3 promoters, while on the contralateral side, NSE and LAT vectors produced moderately higher levels compared to the hGUSB vector, suggesting that some neuronal transport had occurred. Overall, all tested promoters sustained stable and high expression of GUSB [55].

More precisely defined regulatory elements of the aforementioned herpesvirus LAT gene were used in a more recent work. Maturana et al. identified two latency-associated promoters (LAPs) in the genome of the pseudorabies virus. The authors compared elements LAP1, LAP2 and LAP1_2 (LAP1 and LAP2 put together in tandem) in AAV vectors expressing the mCherry reporter. The vectors were tested in neuronal culture and in vivo (a whole-CNS transduction via a retro-orbital injection in mice). The promoters showed different expression efficiencies (LAP2 being the strongest) but similar time courses of expression changes. In culture, the highest level of mCherry fluorescence was at 28 days post-infection (dpi) for all three promoters. Between 38 and 90 dpi, all promoters showed a fluorescence decrease, most likely due to the senescence of neurons after being cultured for so long. In vivo, all three vectors exhibited widespread and long-term mCherry expression throughout the brain. Co-immunostaining for mCherry and neuronal or glial markers revealed that the transgene expression is predominant in neurons. For all promoters, mCherry fluorescence was stable and not significantly different between 30 and 190 dpi time points in the cortex, dentate gyrus, striatum, and cerebellum. Abundant mCherry expression was also observed at 190 dpi in the dorsal and ventral horns of the spinal cord at the cervical, thoracic, and lumbar levels [128].

In some cases, life-long transgene expression seems to be possible for gene therapy of neuropathologies. There are a number of publications confirming this for Parkinson’s disease (PD). Sehara et al. used cynomolgus macaques (*Macaca fascicularis*) to study the gene therapy approach in PD. In this work, AAV-2 vectors with the CMV promoter with human growth hormone intron were used. After the pharmacological induction of PD-like symptoms, the monkeys were treated with the mix of three viruses expressing enzymes necessary for the synthesis of dopamine (tyrosine hydroxylase (TH), aromatic L-amino acid decarboxylase (AADC) and guanosine triphosphate cyclohydrolase I (GCH)) via an injection into the left putamen; in the right side, PBS was injected as a control. Two weeks post-procedure, the monkeys exhibited distinct behavioral recovery, and this remained stable for 15 years. One monkey was euthanized at 15 years and 5 months after the gene therapy, and the histological assessment of the brain was performed. Strong immunoreactivity for TH, AADC, and GCH was observed in the left putamen, while in the PBS-treated right putamen, the expressions of all three enzymes were profoundly lower than in the control (healthy) monkey. Immunofluorescence staining revealed the co-expression of these enzyme genes in the same cells. The transduced region occupied 91% of the left postcommissural putamen. The majority of transduced cells were NeuN+ and appeared to have a morphology typical of medium spiny neurons. Estimation with qPCR showed 81 copies of the vector genome per cell being present in these neurons [117,179]. These results are in agreement with the observations of another study in which AAV2-hAADC-treated parkinsonian monkeys were observed for 8 years [174].

Another example of a pathology that may be treated with long-term viral transgene expression is mucopolysaccharidosis type I, a multisystem genetic disorder with progressive cognitive impairment beginning in early childhood. It is caused by mutations of the *IDUA* gene encoding the lysosomal enzyme α-l-iduronidase. Hordeaux et al. investigated the long-term performance of the AAV9 virus expressing human *IDUA* under the CB7 promoter (similar to CBA). The virus was injected intrathecally to 1-month-old rhesus monkeys, with half of the animals tolerized to the human transgene at birth. Sustained expression of the transgene for almost 4 years was reported in all animals. Both tolerized and non-tolerized animals maintained transgene expression as measured by the immunohistochemical analysis of brain tissue. However, the presence of antibodies in the non-tolerized animals led to a loss of measurable levels of secreted enzyme in cerebrospinal fluid [79]. This result emphasizes the importance of considering possible immune responses that may prevent successful long-term gene therapy even if the protein expression is maintained.

## 4. Non-Promoter Cis-Regulatory Elements Used in Viral Vectors for Epilepsy Gene Therapy

### 4.1. Using Enhancers for Specific Targeting of Neuronal Subtypes

It is estimated that in the human genome, there are ~24,000 genes and ~46,000 promoters, but millions of putative enhancers, which implies that the same gene may be expressed in different cell types by using different sets of enhancers acting upon the same promoter [180].

One of the problems with choosing cell type-specific promoters for gene therapy is that the relatively small payload size of AAVs puts most native promoters out of reach. However, most enhancers are much smaller than promoters. Another advantage is that enhancers are highly conserved between species, so potentially the same regulatory element may be used for manipulating gene expression in different organisms [43]. Enhancer-based targeting proved to be particularly useful for GABAergic interneurons that are important for epilepsy therapy.

Dimidschstein et al. [56], in search of smaller regulatory elements specific to GABAergic interneurons, considered their embryonic stage. The distal-less homeobox 5 and 6 (*Dlx5/6*) genes are specifically expressed by all forebrain GABAergic interneurons during embryonic development. These genes share a short (400 bp) mDlx enhancer sequence, highly conserved across vertebrates. The authors used this enhancer to restrict the expression of transgenes to GABAergic interneurons after an AAV transduction. For experiments in vitro, the calcium indicator GCaMP6f was used as a reporter, and primary neuronal cultures from the mouse cortex were infected with rAAV-mDlx-GCaMP6f. About 95% of cells that expressed GCaMP6f also expressed the pan-interneuron marker GAD67. GCaMP6f expression was high enough to perform calcium imaging. For experiments in vivo, mice were injected with rAAV-mDlx-GFP in the somatosensory cortex (S1), hippocampus or striatum. Sparse but strong GFP labeling was observed. GFP+ cells coexpressed markers of interneurons in their corresponding region, namely, Nkx2.1 in the striatum and PV, SST or VIP in S1. In another experiment, a virus expressing the designer receptor Gq-DREADD under the control of mDlx was injected into the somatosensory cortex of mice. Upon the bath application of clozapine-*N*-oxide (CNO), the ligand of Gq-DREADD, all interneurons expressing Gq-DREADD showed membrane depolarization. Thus, rAAV-mDlx-Gq transduction caused the functional and interneuron-specific expression of Gq-DREADD, allowing the researchers to manipulate the activity of interneurons with CNO. To investigate whether the mDlx enhancer could also drive expression in interneurons not derived from the Dlx5/6 lineage, newborn mouse pups were injected with rAAV-mDlx-GFP in the lumbar region of the spinal cord. Interestingly, in these mice, GFP expression was restricted to the dorsal horns, and these GFP+ cells did not express Pax2, the marker of GABAergic and glycinergic interneurons of the dorsal horns. So, in contrast to the forebrain, rAAV-mDlx-GFP does not target inhibitory neurons in the spinal cord, and the population of spinal neurons targeted by this construct remains to be characterized. The authors also showed that mDlx, when included in an AAV expression cassette, drives transgene expression selectively in brain GABAergic interneurons in zebra finches, ferrets, gerbils and marmosets (in experiments in vivo, with different reporter genes and injection sites), as well as in human cells (interneurons derived from iPSCs and human embryonic stem cells, rAAV-mDlx-GFP). Overall, these results show that the mDlx enhancer can be used in the context of AAVs to broadly and specifically target and manipulate interneurons from the Dlx5/6 lineage across a broad range of species [56].

The same group then continued the search for interneuron-specific enhancers, but focused more on possible therapeutic applications. Vormstein-Schneider et al. investigated the regulatory landscape of the *SCN1A* gene, whose expression is largely restricted to PV+ cortical interneurons. The candidate enhancers were inserted into an AAV vector containing a minimal promoter and dTomato reporter, and the viruses were injected into the retro-orbital sinus of adult mice. Comparing the localizations of dTomato and interneuron subtype markers, the authors identified *SCN1A* enhancers selective for PV+ and VIP+ interneurons. In further experiments, the PV-selective enhancer E2 was successfully used in mice with different transgenes—the presynaptic reporter synaptophysin, the calcium sensor GCaMP6f, different chemogenetic receptors and the red-shifted opsin C1V1. Notably, even though a systemic injection of the virus was used, virtually no reporter expression driven by E2 was observed outside the brain. AAV viruses with E2 enhancer and a fluorescent reporter were also tested in rats, marmosets and macaques. These viruses were able to target PV+ cortical interneurons with approximately 90% specificity across all three species. Additionally, the cell type specificity of an AAV virus with E2 enhancer was confirmed ex vivo in human brain tissue (subiculum or medial temporal cortex), obtained during surgical resection [17].

Nair et al. used brain region-specific enhancers for generating transgenic mice. The authors showed that combining enhancers uniquely active in particular brain regions with a heterologous minimal promoter (the approach they called “enhancer-driven gene expression”, or EDGE) achieves better anatomical specificity compared with the use of native promoters. Later, this group extended this strategy to the generation of AAV vectors. They chose a mutated minimal CMV promoter and a few enhancers previously shown to be specific to particular subsets of neurons in the entorhinal cortex (EC). The experiment was designed in such a way that the enhancer was the only factor ensuring cell type specificity. The AAV serotype used in the study may cross the BBB, so the viruses were administered i.v. For the medial entorhinal cortex enhancer MEC13-53, it was confirmed that MEC13-53 EDGE rAAV expresses the reporter (GFP) specifically in MEC layer II (LII) stellate cells in WT mice, which recapitulates the cell-type specificity previously shown in the MEC13-53 EDGE transgenic crosses. Within LII, there are two major classes of excitatory principal neurons, reelin (RE)+ stellate cells and CB+ pyramidal cells. Immunohistochemical analysis showed that 96% of GFP+ cells in LII were RE+ (for the control CMV-rAAV virus, only 34% of GFP+ LII cells were RE+). There was also sparse GFP expression by MEC13-53 EDGE rAAV in other brain regions, mostly the regions expressing the transgene in MEC13-53 transgenic lines. Moreover, a similar expression pattern for this virus was found in WT rats, even though the enhancer-containing regulatory part of the vector was designed using the mouse genome. The authors also tested several other enhancers with previously shown specificity in MEC neurons. Two of these, when used in AAV vectors, recapitulated the specificity of their corresponding EDGE lines. Namely, MEC13-104 rAAV showed the relatively sparse labeling of a subset of LIII neurons, while LEC13-8 rAAV was mainly LIII-specific. Not all enhancers showed the same specificity when used in viruses as when inserted in the genome; this may be due to transgenic lines possibly having highly specific expression patterns only because of insertional effects [43].

Mich et al. used a dual vector approach to arrange the ectopic expression of human *SCN1A* in GABAergic neurons of mice. They generated a split-intein fusion form of *SCN1A* to circumvent AAV packaging limitations, so the functional protein would be assembled from the products expressed by two different constructs. The author used the CMV promoter, and to specifically transduce telencephalic GABAergic interneurons, they added the optimized hDLXI5/6i enhancer (DLX2.0). They confirmed the specificity of transduction by visualizing GABA+ cells. Infection by the two constructs expressing halves of *SCN1A* was sufficient to rescue DS symptoms in mouse models. Remarkably, after the virus’ injection at the neonatal stage, all treated *Scn1a^fl/+^;Meox2-Cre DS* mice survived beyond postnatal day 70, while untreated animals of this line usually demonstrate ∼50% mortality by P70. The authors also showed that similar constructs but without DLX2.0 and with the SYN promoter caused the non-selective neuronal expression of *SCN1A*, and this caused pre-weaning mortality in mice [181]. However, some other researchers noted that the Dlx elements might not be optimal for vectors used for treating DS because these elements are operative only in forebrain regions, while Na_V_1.1 channels are found throughout the CNS, including more caudal brain regions such as the cerebellum and brainstem [95].

Overall, using short cell type-specific enhancers seems to be a very promising strategy for designing gene therapy vectors, especially considering that enhancers are usually conserved between rodents and primates. It was already shown that the selected enhancers may successfully drive the expression of the *SCN1A* transgene selectively in inhibitory interneurons, making it potentially usable for treating DS. However, this approach is still new, and the studies cited above lack some details important for the clinical application, the first of which being the duration of stable expression driven by these enhancers (although in [181], the therapeutic effect of an enhancer-driven transgene was maintained for at least 70 days).

### 4.2. miRNA Binding Sites Used to Increase Cell Type Specificity

An additional way to control what cell types express the transgene is to take advantage of endogenous posttranscriptional RNA interference mechanisms. miRNAs bind mRNA transcripts and target them for degradation, and a large fraction of miRNAs are tissue-specific and/or cell type-enriched [182,183]. For gene therapy, this allows vector designers to use a promoter active in multiple cell types and combine it with cell type-specific miRNA binding sites to “de-target” expression in cells where it is not desired. Binding sites for miRNAs are small, can be combined, and are robust in their ability to restrict expression. Notably, miRNAs are particularly enriched in the mammalian brain compared to other organs [40,150].

In the context of CNS, most of the commonly used viral vectors are highly neurotropic, and adding miRNA binding sites would allow for making transgene expression restricted to glial cells with little modifications to these vectors. Colin et al. added four copies of the neuron-specific miR-124 target sequence at the 3′-end of the LacZ reporter in an LV vector with PGK universal promoter. Viruses were injected into the striatum of adult mice, and immunostaining for LacZ, NeuN and S100β (glial marker) was used to determine the cell type of transduced cells. miR-124 binding sites allowed the authors to de-target transgene expression in the murine brain from mostly neurons to mostly astrocytes. In the same study, it was also found that pseudotyping (using an envelope based on a different kind of virus) with Mokola virus proteins shifts the tropism of LV vectors toward astrocytes. The authors then successfully used this combined strategy, Mokola pseudotyping and miR-124-dependent detargeting, to express other transgenes, GFP, glial glutamate transporter GLAST, or an shRNA against GLAST, selectively in astrocytes [184].

Keaveney et al. utilized miRNA binding sites to selectively target GABAergic interneurons. This microRNA-guided neuron tag was abbreviated “GABA mAGNET”. The LV-GABA-mAGNET vector contained the SYN promoter, the EGFP reporter, and multiple copies of miR128 and miR221 target sites in the 3′-UTR. miR128 and miR221 are miRNAs with both high expression levels in excitatory neurons and the highest ratios of expression between excitatory and inhibitory neurons. The SYN promoter was chosen for its moderate strength; a stronger promoter could possibly overcome the effects of the posttranscriptional regulation by endogenous neural miRNAs in off-target cells. The virus was injected into the mouse cortex. Transgene expression in inhibitory or excitatory neurons was estimated by the colocalization of EGFP with immunostaining for either GABA or CaMKIIα, respectively. LV-GABA-mAGNET predominantly labeled interneurons, exhibiting 91% target selectivity. It was then further improved by packing the same construct into an AAV vector (more specifically, AAV2 genomic backbone pseudotyped with AAV9 capsid protein). AAV-GABA-mAGNET achieved 98% targeting selectivity for cortical interneurons, which is on a level with the cell-type targeting standards set by transgenic animals. Immunohistochemistry for PV and CR showed that EGFP+ cells expressed these markers in the proportions corresponding to their expected distributions in mouse cortex. Thus, AAV-GABA-mAGNET is highly specific for all cortical interneurons without a labeling bias between interneuron types. The viruses designed for mice also showed preferential expression in interneurons in rat brain. In the rat cortex, LV-GABA-mAGNET and AAV-GABA-mAGNET achieved 74% and 82% interneuron targeting selectivity, respectively. Thus, miRNA-based gene-targeting exhibits some cross-species functionality, even though species-specific miRNA profile data would be necessary to optimize vector design [150].

In the study cited above, where transgene expression restricted to inhibitory interneurons was achieved in the mouse model of DS using *GAD1* promoter, Tanenhaus et al. also added eight target motifs for miR128 and miR221 to the 3′-UTR of the construct to increase cell type specificity [14].

### 4.3. WPRE Elements

Elements that increase protein expression at the posttranscriptional level were found in the genome of the woodchuck hepatitis virus, and their use for gene therapy was proposed by Loeb et al. in 1999. Woodchuck hepatitis virus posttranscriptional regulatory element (WPRE) is a cis-acting RNA sequence that facilitates mRNA processing and export to cytoplasm. In intronless viral transcripts, WPRE and its homologs in other viruses apparently act similarly to introns in stimulating processing. Loeb et al. added WPRE to an AAV vector with the CMV promoter and eGFP reporter gene, and confirmed that WPRE increases transgene expression in transduced HEK293 cells and primary human fibroblasts [185].

In 2002, Klein et al. first demonstrated that WPRE increases transgene expression when coupled with the CBA promoter. In this study, the authors compared a few different promoters for the neuronal expression of transgenes. They used AAV2 vectors and the GFP reporter. It was discovered that adding the WPRE element into a construct with the CBA promoter induced greater expression levels in primary neuronal cultures and in the hippocampus. At 1 month after the intrahippocampal injection, Western blotting showed 11.4-fold greater GFP expression in the hippocampus of rats injected with the WPRE-containing virus, as compared to the similar virus without WPRE. Microscopy showed that the WPRE-containing virus induced GFP expression in more cells in the hippocampal tissue, including even some cells in the contralateral (uninjected) side [123].

In different studies, WPRE was found to increase transgene expression driven by CMV, PPE (preproenkephalin), NSE or PDGF promoters (in vitro and/or in vivo) [40,132,186]. However, expression levels driven by intron-containing EF1α or CAG promoters are not much increased by including WPRE [187].

Very recently, Peterman et al., who measured both the transcription and translation rates of plasmids with different elements, showed that adding the WPRE sequence leads to the greater accumulation of mRNA transcripts, but these transcripts are translated at a lower rate. Strong promoters (CAG, EF1α) are already approaching the maximum transcriptional output, so WPRE is not effective when coupled with them, and even decreases the protein production. An aberrant localization of transcripts from constructs with WPRE was also demonstrated; they are not uniformly distributed in the cytoplasm, but form puncta, which may explain the hindered translation. These new observations cast doubt on the conventional theory that WPRE increases protein expression at the posttranscriptional level. However, in this work, WPRE’s effects were not studied in non-integrating viral vectors. Indeed, WPRE action may be very different depending on the transgene delivery method; in a separate experiment with the site-specific integration of different constructs into the HEK293T genome, WPRE’s presence significantly increased protein levels for the CMV and EFS promoters despite minimal effects at the mRNA level. It should also be noted that in this paper, WPRE was used in place of a PAS sequence, while in other studies, the role of WPRE in polyadenylation is not mentioned, and a separate PAS is normally added to expression constructs with WPRE [42].

Among the studies that used WPRE in viral vectors expressing transgenes related to epilepsy are [27,94,95,96,139], which are cited above. In addition, the present review cites some papers wherein WPREs were used to design cell type-specific vectors [150,184].

WPRE is a 600 bp-long tripartite element. Choi et al. were able to minimize its size. As an initial template, they used a strong expression cassette, which contains the CaMKII promoter, the EGFP coding sequence, WPRE, and the bovine growth hormone PAS (bGHpA). In this expression cassette called “CaMKII, WPRE, bGHpA” (CWB), the authors replaced WPRE with shorter elements, and examined the transgene expression efficacy of the modified cassettes. They performed a similar optimization with bGHpA, replacing it with shorter PAS sequences. It was found that the size of WPRE may be markedly reduced (to 247 bp) without significantly decreasing the expression level of the transgene. For the cassette with the shortened WPRE variant, the EGFP expression in the cultured hippocampal neurons transduced with an AAV1 vector was 83.4% of the expression level for the original CWB cassette. Thus, the optimized vector design allows one to package larger transgenes into AAVs without compromising expression efficiency [188].

Table 5 summarizes types of non-promoter elements regulating transgene expression that are described in the present review.

As a conclusion, in addition to promoters, there are diverse regulatory elements that may help to achieve the desired properties of an expression vector—to significantly increase the transgene expression level or to fine-tune the expression to selected cell types. Non-promoter genetic elements may be used in combination with any promoter, even though some combinations would be less effective than the others (for example, WPRE does not much increase expression driven by intron-containing promoters).

## 5. Future Considerations: Search for New Cis-Regulatory Elements with State-of-the-Art Methods

Even though some studies of gene therapy for epilepsy are already in the first stages of clinical trials, it is necessary to continue the search for new regulatory elements that may be used for either the more precise targeting of cell types involved in epileptogenesis or located near epileptic foci, or for the more fine-tuned control of gene expression levels. In the following sections, we discuss some methods used for this search, including big data approaches.

### 5.1. Screening for Novel Promoters

Some features of promoters are detectable in the DNA sequence itself. In vertebrate genomes, these are CpG islands close to the TSS, the presence of typical core and proximal promoter elements, and TF binding sites. GC content and the structural DNA properties that may be calculated based on its sequence (like DNA denaturation values or bendability) can be used to predict core promoters [53]. Since motifs of the core promoter elements are highly degenerate, analyses of sequence conservation between different species can greatly improve the identification of these elements [189]. However, the epigenetic properties of genome regions provide much more information useful to identifying functional promoters. It should be noted that the methods described below only predict active promoters of genes that are transcribed in the studied cells; different tissues, cell types, development stages, etc., have different sets of active promoters.

The introduction of chromatin immunoprecipitation sequencing (ChIP-Seq) allowed us to map protein–DNA interactions across entire genomes, including the identification of TF binding sites that do not have previously known motifs. In ChIP experiments, an immune reagent specific for a chosen DNA binding factor is used to enrich target DNA sites to which the factor was bound in the living cell. The enriched DNA sites are then identified [190].

CpG dinucleotides at promoters are almost always unmethylated. Conversely, intragenic CpG islands are often methylated [191]. The methylation of CpG islands in promoters is also generally considered a hallmark of tissue-specific promoter silencing. Various high-throughput methods have been invented for the large-scale detection of DNA methylation events by sequencing or microarrays, including the use of methylation-sensitive restriction enzyme digestion, immunoprecipitation, affinity capture, and the bisulfite conversion of unmethylated cytosines to uracils. These methods allow the prediction of active promoters in different cell types. However, the integration of gene expression data and global DNA methylation profiles showed that, while there is a general negative correlation between promoter methylation and gene expression level, there is still a substantial overlap in the distributions of promoter methylation levels between genes with low versus high expressions [192]. Histone methylation and acetylation are other important epigenetic modifications that may be used to predict promoters. Generally, high levels of histone acetylation and H3K4 methylation are detected in promoter regions of active genes; high levels of H3K4me1, H3K4me2, and H3K4me3 are detected surrounding TSSs. To assess histone modification on a genome-wide scale, ChIP-Seq with antibodies against modified histones is used [193].

Promoter activation is accompanied by open chromatin. It may be mapped by assay for transposase-accessible chromatin using sequencing (ATAC-Seq). This method probes DNA accessibility with hyperactive Tn5 transposase, which inserts sequencing adapters into accessible regions of chromatin [194]. Additionally, DNase I hypersensitive sites sequencing (DNase-Seq) allows one to identify sites of open chromatin presumably interacting with TFs. DNase I selectively digests nucleosome-depleted DNA, whereas DNA regions tightly wrapped in nucleosomes are more resistant [195].

Since pol II promoters include the TSS and are partially transcribed, RNA-Seq data are also useful for locating TSS and their corresponding promoters. In cap analysis gene expression (CAGE), short (approximately 20 nucleotide) sequence tags originating from the 5′ end of mRNAs are sequenced to identify transcription events on a genome-wide scale. CAGE relies on a cap-trapper system to capture full-length mRNAs while avoiding rRNA and tRNA transcripts. CAGE identifies the location of each TSS in addition to the expression level of the corresponding transcript. This makes it possible to search the promoter region surrounding each CAGE-defined TSS for potential TF binding sites [196].

Mammalian promoters are generally large and complex, with cis-regulatory modules (CRMs) dispersed throughout the gene. Identifying CRMs and predicting their function remains a major challenge. De Leeuw et al. developed a high-throughput pipeline that includes human genome screening for CRMs, the design of artificial candidate MiniPromoters (short cell type-specific promoters that have CRMs of a given gene combined together) and their rapid testing in vivo. This group previously designed and tested MiniPromoters for use in transgenic mice; typical candidate promoters were ∼4 kb, consisting of a PROM (a region containing the core promoter, 1.1 kb at the TSS on average) and multiple CRMs (average length 900 bp), typically repositioned upstream of the PROM. However, this is too long for using in AAV vectors. Additionally, MiniPromoters with confirmed cell type-specific expression in knock-in animals may not function as well when introduced into AAVs because multiple virus copies may broaden their expression, and because developmentally-established epigenetic marks may not be acquired by a promoter introduced after birth. In this study, some MiniPromoters specific for different brain or eye cell types and already used in transgenic mice were unaltered, while others were shortened to 2.5 kb or smaller to use them in AAVs. A few completely new candidates were also tested. In total, 19 candidate promoters were tested in AAV2/9 with improved Cre recombinase (icre, used in a Cre-reporter mouse line) or GFP reporter genes. Viruses were injected i.v. to mouse pups. All candidates showed consistent restricted expression (expression pattern related to their corresponding source gene) in different regions of the brain or the eye [197]. Later, the same group developed an approach to simplify the custom design of MiniPromoters in silico. It involves identifying candidate CRMs of a gene of interest based on their conservation and epigenetic properties, and combining a subset of the resulting CRMs into a MiniPromoter. Fornes et al. made OnTarget, an online application for designing MiniPromoters. Users need to specify a gene or genomic coordinates on which to focus the identification of CRMs. OnTarget identifies candidate CRMs and suggests MiniPromoter designs. As an example, the authors used OnTarget to design two MiniPromoters based on the regulatory elements of genes *ADORA2A* and *PITX3*, expressed specifically in striatal D2-type medium spiny neurons and dopaminergic neurons of the substantia nigra, respectively. These cell populations are relevant to Parkinson’s disease. The designed promoters were then characterized in vivo. For comparison, two MiniPromoters designed manually using the same genes were also tested. Each MiniPromoter was cloned into an AAV vector with an EmGFP reporter and packaged into AAV9 or rAAV-PHP.B capsids for widespread brain transduction. Adult mice or pups (P4) were injected i.v. with the viruses. For both genes, the promoters designed using OnTarget showed both strong and specific expression of the reporter in the target cells, while the manually designed promoters were not effective [58].

### 5.2. Novel Enhancers Search

In addition to designing new promoters, new methods of making enhancers usable for gene therapy are being developed. Enhancers are usually smaller than promoters and may be more selective for specific cell types. However, artificial genetic constructs containing enhancers started being designed only recently, because newly developed sequencing techniques made it easier to identify naturally occurring enhancers located far from their corresponding genes. In addition, systematic enhancer discovery has been accelerated by the advent of technologies allowing for transcriptomic and epigenetic studies at single-cell resolution [17].

Many screening methods assessing epigenetic properties or TF binding sites on a genome-wide scale described in the previous section, including ChIP-Seq, ATAC-seq, and DNase-Seq, may be used for locating enhancers as well as promoters. Additionally, there are relatively recently developed methods for detecting chromatin interactions allowing one to detect promoter–enhancer interaction events, like high-throughput chromosome conformation capture (Hi-C), chromosome conformation capture coupled with sequencing (4C), or chromatin interaction analysis by paired-end tag sequencing (ChIA-PET) [198]. These methods are modified and improved versions of chromatin conformation capture (3C) that only allows one to study selected promoter–enhancer interactions. The original 3C method means the formaldehyde cross-linking of proteins to proteins and to DNA, the subsequent digestion of cross-linked DNA by restriction enzymes, the ligation of cross-linked DNA fragments, and the assessment of selected ligation junctions by PCR [199]. Single-cell ATAC-seq (scATAC-seq) is a method that combines the single-cell approach with ATAC-seq. This allows one to identify accessible chromatin in different types of cells [200].

To reliably identify enhancers of a given gene, chromatin accessibility data may be combined with other factors, like proximity of non-coding elements to the gene TSS and the conservation of these elements between species. It was confirmed that this in silico approach allows one to select a few putative enhancers for a particular gene. These candidate enhancers can then be inserted into viral vectors and tested in vivo. Using this strategy, Vormstein-Schneider et al. successfully identified three enhancers driving *SCN1A* expression in different neuron subtypes after testing 10 candidates. They also tested 25 enhancer candidates in the vicinity of seven other genes with enriched expression in the PV+ interneurons, and four of the candidates displayed >90% selectivity for these interneurons [17].

Blankvoort et al. used enhancer ChIP-seq to identify enhancers specific for different cortical subregions microdissected from mouse brain. They identified a total of 59,372 putative enhancers, with 3740 of these being specific to particular cortical subregions. Of these, 10 were tested for their efficiency and specificity in vivo by generating transgenic mice expressing a reporter under control of a minimal promoter coupled with the selected enhancer. The visualization of expression patterns in these animals confirmed that it is possible to obtain highly specific targeted gene expression using regionally specific enhancers associated with non-specific genes (genes expressed throughout the brain) [180]. Later, Nair et al. used some of these enhancers to drive the region-specific expression of transgenes introduced by AAVs [43]; this paper is described in Section 4.1.

Very recently, Furlanis et al. [144] performed a systematic analysis of single-cell genomic data to identify enhancer candidates for different cortical interneuron subtypes. The sequences for the selected enhancer candidates were cloned into an AAV construct with a minimal promoter and dTomato reporter. The authors tested different administration routes (i.c.v., retro-orbital i.v. or intracranial stereotactic injection) in mice of different ages (from newborn pups to 8-week-old adults). The specificity of the reporter expression for the intended target population was determined by the colocalization of dTomato and cell type-specific markers in brain slices. After the in vivo experiments with dozens of enhancer candidates, the authors identified the seven best enhancers that are highly specific for distinct neuronal populations and can be used in AAV vectors. Additional experiments confirmed that these enhancer elements may be successfully paired with different transgenes including fluorescent reporters, calcium sensors and channelrhodopsins. The targeted cell types include all main types of GABAergic interneurons—VIP+, Lamp5+, PV+ (distinct enhancers were designed for basket cells and chandelier cells) and SST+ (different enhancers for non-Martinotti and Martinotti cells). Additionally, the authors identified an enhancer specific for striatal cholinergic neurons and confirmed its specificity in vivo in a similar way. The identified enhancer sequences are conserved between mice and primates. To test some of the selected enhancers in primates, AAVs carrying expression cassettes were delivered intracranially into rhesus macaque cortex, hippocampus, or striatum. The combined immunohistochemistry and electrophysiology data obtained in experiments with macaques strongly support selectivity for the intended neuronal subpopulations of the tested enhancer-based AAVs. Moreover, the VIP interneuron-specific enhancer was additionally validated in human brain tissue surgically resected from the temporal lobes of two patients. The authors propose the identified enhancers as valuable tools to target and manipulate distinct populations of neurons for studying them. Importantly, the identified cell type-specific enhancers showed highly variable labeling in mice when injected at different ages and via different injection routes, and the optimal delivery protocols for different enhancers were different. The authors also note that the orientation of an enhancer can be critical for its function and that the relationships between expression level and selectivity may vary significantly for different transgenes [144].

### 5.3. Methods of Validation for Putative Regulatory Elements

Finding a strong and specific cis-regulatory element that also has a moderate size is still a complicated task because the methods currently used for the prediction of functional promoter and enhancer sequences are not very accurate, and multiple candidate sequences found in the genome need to be tested for their activity when inserted in expression vectors to validate the predictions.

Traditionally, time-consuming reporter assays have been used to directly test if a DNA sequence acts as a functional promoter or enhancer. To enable screens on a larger scale, the massive parallel reporter assay (MPRA) method was developed. MPRA allows one to test multiple regulatory elements on a small number of biological samples by using barcodes. A candidate promoter sequence is linked to a reporter gene whose 3′-UTR includes a unique sequence tag (barcode). The reporter vectors are introduced into cell lines as plasmids, total RNA is extracted from the culture and sequenced, and the reporter gene expression for each barcode is estimated (relative to the amount of its plasmid DNA). MPRA for enhancer candidates is similar, except the reporter gene sequence also contains a minimal promoter in this case. Cis-elements that are able to increase gene expression would increase the number of reads for the corresponding barcode [201,202,203]. This method may also be used in vivo if the MPRA constructs are packaged into AAV capsids [204]. Zhao et al. developed a modification of the MPRA assay, single-cell massively parallel reporter assay (scMPRA), to measure the activity of libraries of cis-regulatory sequences (CRSs) across multiple cell types simultaneously. This method combines MPRA and single-cell RNA-seq, and allows one to test multiple promoters and their mutant variations using a limited number of animals and select the most specific promoters for a particular cell type. The authors tested multiple regulatory elements in cell line models and mouse retinal tissue ex vivo. They have successfully identified mutant versions of the *Gnb3* promoter specific for different types of retinal cells [205].

The self-transcribing active regulatory region sequencing (STARR-seq) technique is an upgrade of the MPRA approach for enhancer screening that incorporates the tested genomic fragments downstream of a minimal promoter [206]. Because enhancers can function independently of their relative positions, active enhancers, when put downstream of a promoter, simply facilitate their own transcription. Moreover, each enhancer’s strength is reflected by its abundance among cellular RNAs [207]. This eliminates the need for barcodes, simplifies library generation and enables the scaling to genome-wide screens of entire mammalian genomes. Chan et al. performed a massive enhancer screening using STARR-seq in vivo. They used AAV vectors to deliver a highly complex screening library in mouse brains. The library contained ∼700 bp long genomic fragments corresponding to regions covering a selected set of genes with strong and constitutive expression in the brain. The library in total spanned about 3 Mb of the mouse genome. Each fragment was cloned into a screening vector (scAAV with AAV2 ITRs and a minimal promoter derived from plasmid GL4.26) that was packaged into AAV8 capsid. Viruses were injected in 12 different sites covering a major part of the dorsal neocortex. After the viral expression, RNA from the cortical tissue surrounding the injection sites was isolated and sequenced. Hundreds of thousands of different RNA sequences were assessed in each mouse; 483 sequences with enhancer activity were identified (each was present in all five mice), including sequences that were not predicted by DNA accessibility or histone marks in the cortex (for some of these fragments, these markers have been reported in non-brain organs). The nine selected candidate enhancers were tested with traditional in vivo fluorescence reporter assays. Most of them showed an expression level comparable with that of the CMV promoter, with one candidate providing much higher expression. Two of the candidates were studied in more detail; experiments with co-labeling with different antibodies showed that they drive expression in neurons, but with different subtype preferences, and are not active in oligodendrocytes. Notably, one of these drove expression only in a very sparse subset of neurons (mostly PV+ interneurons, based on immunohistochemistry results), indicating that the AAV-STARR-seq screen has a remarkably high sensitivity to detecting signals that stem from only a minor fraction of cells [206].

## 6. Conclusions

We have discussed multiple examples of regulatory elements designed for targeting specific cell types and/or cell states. In the case of epilepsy, one of the approaches to gene therapy is to make the expression of therapeutic transgenes incredibly fine-tuned, specific for a particular neuronal subtype and/or responsive to the frequency of firing in these cells. Single-cell transcriptomics data identified specific types of glutamatergic principal neurons and GABAergic interneurons as cells most affected by epilepsy [16]. The works cited above explore different approaches to targeting principal neurons, interneurons and astrocytes.

Designing regulatory elements capable of cell type-specific expression control is a very hard task, especially considering the limited payload size of the most commonly used viral vectors. Currently, the most common approach is to use AAV-based vectors that allow packaging of about 4.4 kb of non-viral DNA (and only about 2.2 kb for scAAVs that have some advantages). It becomes increasingly clear that a rational design of short and specific promoters and other regulatory elements should benefit from new data provided by epigenomics and transcriptomics studies. New techniques made it possible to reliably identify putative promoters and enhancers on a genome-wide scale, and screen thousands of predicted candidates in parallel in functional tests.

In many cases, a short specific enhancer combined with a minimal promoter would take less valuable space in the vector than a cell type-specific promoter. However, this approach is still relatively new, and many aspects regarding enhancers in such vectors are not yet studied well enough—for example, the time course of their activity. There are also examples of enhancers targeting unexpected cell populations in animal experiments, like unidentified neurons in the spinal cord in [56]. Importantly, endogenous enhancer activity may depend on the cell state, so additional experiments are necessary to test transgene expression level and cell type specificity for enhancer-driven genetic constructs during long time periods and/or in different neuron states (including after a seizure). The latter is also true for any other regulatory elements proposed for gene therapy.

Different regulatory elements may be combined together to generate new promoters with unique properties. Some examples of promoters made using these data are in the papers [14,58,59,101,197] described above.

In the past decade, methods of genome-wide screening for putative promoters and enhancers (like ChIP-Seq, ATAC-Seq, scATAC-Seq, DNAse-Seq, Hi-C and others), and more importantly, methods for the massive parallel validation of the predicted sequences (like MPRA or STARR-Seq) described in Section 5.1, Section 5.2 and Section 5.3 became widely available. These methods are very useful for generating new building blocks for expression vectors tailored for specific applications. The cell specificity of a viral vector may also be increased with additional regulatory elements like binding sites for endogenous cell type-specific miRNAs [14,30,40,184]. Modifications of the viral capsid [32,208] or pseudotyping [184,209] should also be considered, and modern screening methods allow the rapid generation and selection of capsids with desired properties [30,210,211].

In some cases, it is possible to increase the cell type specificity of transgene delivery using a combination of two viruses (not unlike combinations used for tissue-specific gene knockout in transgenic mice). If there is a promoter active in a few cell types, the transgene of interest may be floxed and inverted within the vector, and a second virus expressing Cre recombinase under another specific promoter would make the transgene expression possible only in cells where both promoters are active. This “set intersection” approach was successfully used by Mehta et al. with different combinations of promoters and enhancers to precisely target different subtypes of GABAergic interneurons. In addition to the “set intersection” Cre recombinase-dependent method, a “set difference” method with a tetracycline-dependent promoter in one virus and tetracycline repressor expressed by another virus was also confirmed to be useful. In this case, the coexpression of both vectors in the same cell leads to repressed transgene expression, so it is possible to “subtract” a specific subset of cells from all cells expressing the transgene of interest [146].

Currently, expression vector design may be hindered by the lack of exact and controlled nomenclature for regulatory elements. Analyzing multiple papers, we found different names used for the same promoter and a lack of references to the exact sequence of the elements used. In some cases, it is not clear which species’ genome was the source of a particular element. Even worse, there are often conflicting data about the length of the same element in different sources, and the experimental comparison made in [127] for EF1α promoter has shown that promoters used in different commercially available vectors and labeled with the same name may actually have different transgene expression efficiencies. In the same paper, it is noted that there are at least four different versions of the CMV promoter annotated in different vectors listed in the NCBI Nucleotide database, all with the same name but drastically different lengths. To improve this situation, full sequences of all the expression vectors used in a study should be disclosed in research papers whenever possible.

An important consideration when developing any treatments, including the design of viral vectors for gene therapy, is the limited utility of animal models. For example, the neuropeptide galanin has been intensively studied in rodents as a therapeutic transgene, but these studies yielded scarce translational data [9,18]. An experiment on surgically removed epileptic brain tissues showed that galanin had no antiepileptic effect in the human hippocampus, despite confirmed binding with receptors. The authors suggest that the downstream signaling of galanin receptors might be impaired in epileptic tissue [212]. It must be considered that in mice and rats, some CNS properties are fundamentally different than in humans, and choosing potential target genes and their regulatory elements for treating epilepsy with gene therapy should be based on thorough comparisons of brain transcriptomes between species, and between cell types when available. Gene expression patterns that change specifically in epileptic neurons should also be considered, and compared in different species if such data are available. With gene therapy, another possible problem not well predicted by experiments with rodents is the possible immunogenicity of viral, bacterial or chimeric proteins in humans [9,159]. Conversely, the expression of human proteins in model animals (even primates) may sometimes cause an immune reaction leading to inactivation of the foreign protein [79]. However, scientists working in the field of epilepsy have a unique possibility to validate animal studies in live human brain samples, since resected epileptic tissue may be used for neurophysiological experiments. Such ex vivo validation should also be an important step before proceeding to clinical trials, and some of the most recent studies, like [17,74], have already included it.

## Figures and Tables

**Figure 1 cells-14-00236-f001:**
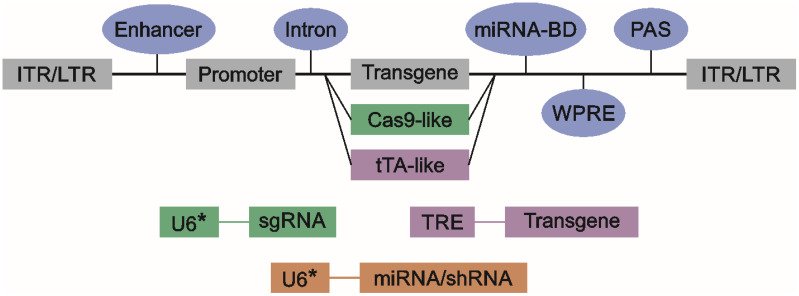
Genetic elements used for the development of gene expression vectors for epilepsy gene therapy. A single vector may contain multiple transgenes, their expression being driven by the same promoter or separate promoters. In gray rectangles are obligatory elements. ITRs or LTRs are viral repeat sequences necessary for the expression of AAV or LV vectors, respectively. In blue ellipses are optional elements that may be added to increase the transgene expression (enhancers, introns, WPRE elements and polyadenylation signal) or to fine-tune it to a specific cell type or specific conditions (miRNA binding sites). PAS is considered obligatory in most studies, but it was demonstrated that its absence does not prevent transgene expression completely. We depict here a few specific types of transgenes separately—small RNAs (sgRNA, miRNA, shRNA), Cas9-like proteins and tTA-like transcription activators/repressors. These transgenes are used to manipulate expression levels of other genes (either in the host genome or transgenes introduced in a viral vector). In green rectangles are elements necessary for Cas9-dependent expression regulation, in purple rectangles are elements necessary for drug-dependent expression regulation. * U6 is the most commonly used promoter for the expression of small RNA species. However, in some studies, other promoters (including pol II promoters) were successfully used to express miRNAs.

**Table 1 cells-14-00236-t001:** Ongoing clinical trials of proposed gene therapies for epilepsy.

Trial Number and Short Title	Expression Vector and Promoter ^1^	Therapeutic Transgene and Its Proposed Function ^2^
NCT06112275. A Clinical Study to Evaluate the Safety and Efficacy of ETX101, an AAV9-Delivered Gene Therapy in Children With SCN1A-positive Dravet Syndrome (Australia Only) [10]	AAV9 (with RE^GABA^ element)	ETX101 expresses the engineered transcription factor eTF^SCN1A^, which increases the transcription of the *SCN1A* gene encoding a subunit of Na_v_1.1, a sodium channel responsible for the generation of action potentials in GABAergic interneurons. Monoallelic *SCN1A* mutations are associated with Dravet syndrome [14].
NCT05419492. A Clinical Study to Evaluate the Safety and Efficacy of ETX101 in Infants and Children with SCN1A-Positive Dravet Syndrome [11]		
NCT04601974. Lentiviral Gene Therapy for Epilepsy [12]	LV	EKC is an optimized version of the voltage-gated potassium channel K_v_1.1. When expressed in excitatory neurons, EKC reduces their excitability and glutamate release [15].
NCT06063850. AMT-260 Gene Therapy Study in Adults with Unilateral Refractory Mesial Temporal Lobe Epilepsy [13]	AAV9 (with hSyn1 promoter)	AMT-260 locally delivers silencing miRNA to target the *GRIK2* gene and suppress aberrantly expressed GluK2-containing kainate receptors.

^1^ The information about regulatory elements in the vectors used in the trial is not disclosed in some studies. ^2^ The information about the proposed mechanisms of the transgene function is not always provided in descriptions of clinical trials, but there are research papers about the same transgenes that we cite here.

**Table 2 cells-14-00236-t002:** Promoters and genes described in the literature about gene therapy for epilepsy.

Promoter	Gene and/or Protein Name	References
CMV	NPY	[62]
	GDNF	[63]
	FGF2	[64]
	*KCNA1* (K_v_1.1 potassium channel subunit)	[65]
	clostridial toxin light chain	[35,66]
	*NMDAR1* cDNA (used in an AAV vaccine to stimulate production of antibodies against NR1 subunit of NMDA receptor)	[67]
	antisense RNA for *NMDAR1*	[68]
HSV α4	Hsp72	[69]
HSV α22	Glut-1 glucose transporter	[70]
ICP10	antiapoptotic viral gene ICP10PK	[71]
ICP0	BDNF	[64]
CBA/CAG	*SCN1A* (Na_V_1.1 sodium channel subunit)	[36]
	preprosomatostatin	[22,72]
	NPY	[73]
	NPY and its receptor Y2	[74]
	full-size NPY or NPY13-36 fragment (with FIB)	[25]
	preprodynorphin	[20]
	GAD67 (glutamate decarboxylase 1)	[75]
	*ASPA* (aspartoacylase)	[76]
	Homer 1a proteins	[77]
	galanin (with FIB)	[26]
	GDNF	[78]
	*IDUA* (α-l-iduronidase)	[79]
EF1α	channelrhodopsin-2 (conjugated to mCherry)	[80]
	dCas9-VP160	[28]
SYN	anti-*Grik2* miRNA	[81]
	hM4Di (inhibitory DREADD)	[82]
NSE	galanin	[83]
	NPY	[84,85]
	NPY and its receptor Y2	[86,87]
	NPY and its receptor Y5	[88]
	*SCN1A*	[37]
CaMKII	EKC (engineered potassium channel, mutant *KCNA1*)	[15,89]
	halorhodopsin	[65]
	hM4Di	[90]
	*KCNN4* (K_Ca_3.1 potassium channel subunit)	[91]
	rtTA (reverse tetracycline transactivator) conjugated to tdTomato	[28]
	dSaCas9-VP64	[92]
	GluCl	[93]
mCamk2a	NPY and its receptor Y2	[94]
truncated *Gad1* promoter	Na_v_β1 sodium channel subunit	[95]
RE^GABA^ (artificial regulatory element containing *GAD1* gene fragments including the promoter)	eTF^SCN1A^ (engineered transcription factor)	[14]
GFAP	GS, EAAT2, miR-ADK	[96]
	NeuroD1	[97]
gfaABC_1_D	antisense RNA for *ADK*	[27]
Promoter with TRE (Dox-Off)	galanin (with or without FIB)	[21]
Promoter with TRE (Tet-Off)	antisense RNA for *NMDAR1*	[68]
Promoter with TRE (Dox-On)	TREK-M potassium channel (mutant *Kcnk2*)	[98]
Promoter with TRE (Tet-Off)	GAD65 (glutamate decarboxylase)	[99]
Promoter with TRE (Tet-On)	dCas9-VP64	[28,100]
c-Fos	EKC, EGFP-T2A-rtTA	[100]
EpiPro (artificial promoter) combined with elements from *Arc* promoter	TREK-M	[101]
GABRA4	*GABRA1* (α1 subunit of GABA_A_ receptor)	[102]
U6	shRNA for *SCN8A*	[103]
	miRNA for a pathogenic *Dnm1* allele	[104,105]
	sgRNA for *Alox5*	[106]
	sgRNA targeting *KCNA1* promoter	[28,100]
	sgRNA for *KCC2*	[92]
	sgRNA targeting *Scn1a* promoter	[29]

**Table 3 cells-14-00236-t003:** Properties of the most important universal promoters described in Section 2.1. We focused on studies wherein expression vectors with these promoters were administered to the CNS.

Name	Design	Promoter Efficiency	Expression Duration	Cell Type Specificity ^1^
CMV	human cytomegalovirus immediate-early enhancer and promoter	Variable, expression in neurons depends on neuronal activity [114,115]	Silenced after 10 weeks in spinal cord [54]	90.6% expression in NeuN+, 9.4% in GFAP+ cells in rat cortex; 92.6% expression in NeuN+, 7.4% of GFAP+ cells in monkey cortex [125]
CBA/ CAG	CMV immediate early enhancer and the chicken β-actin (CBA) gene promoter; may also contain other elements of the CBA gene and a splice acceptor of the rabbit β-globin gene	High [42,108]	Expression maintained at least 6 months after intrastriatal injection [126]	99.5% expression in NeuN+, 0.5% in GFAP+ cells in rat cortex; 85.3% expression in NeuN+, 14.7% of GFAP+ cells in monkey cortex [125]
EF1α	Promoter of the human eukaryotic translation elongation factor 1 alpha 1 (*Eef1a1*) gene	High (similar to CAG in cell lines [108]), but different versions of this promoter used in vectors have different levels of efficiency [127]	Expression in different brain regions maintained at least 190 days after infection [128]	99.9% expression in NeuN+, 0.1% in GFAP+ cells in rat cortex; 92.2% expression in NeuN+, 7.8% in GFAP+ cells in monkey cortex [125]

^1^ NeuN—neuronal marker; GFAP—astrocyte marker used to identify cell types in immunohistochemistry experiments. A small fraction of cells positive for these markers may belong to other cell types.

**Table 4 cells-14-00236-t004:** Properties of cell type-specific promoters described in Section 2.2.

Name	Design	Promoter Efficiency	Expression Duration	Cell Type Specificity ^1^
NSE	Rat *Eno2* (*NSE*) gene promoter	High [132]	25 months [123]	Neurons, with some expression in other cells (NeuN−) and specificity being lower than that of SYN [34,61]
SYN	A fragment of the human *SYN1* gene promoter	High but lower than that of CAG [126]	Stable expression 8 weeks after infection [61]; decreased expression 6 months after infection [126]	Neurons [34] (>96% of transgene-expressing cells were NeuN+ in [61])
CaMKII	Mouse *Camk2a* gene promoter	High [61,152], but low in [126]	At least 120 days [153]	Neurons, mostly excitatory (100% NeuN+, GABA− cells in monkey brain; 93.7% NeuN+, GABA− cells and 6.3% NeuN+, GABA+ cells in rat brain in [125])
Gad1	Truncated mouse *Gad1* gene promoter	Low to moderate	Decreased expression still detectable at the age of 2–3 months after infection at P2 [95]	75–85% of transgene-labeled cells in the mature brain were non-GABAergic neurons [95]
RE^GABA^	Parts of the human *GAD1* gene (enhancer, promoter, 5′-UTR, truncated first intron), miRNA binding sites in 3′-UTR and some other elements	Moderate	At least 27 days [14]	Transgene expression under control of RE^GABA^ was limited to GAD67+, PV+ and SST+ neurons [14]
GFAP	Human *GFAP* or rat *Gfap* gene promoter	Moderate (rat promoter in [132])	At least 3 months (rat promoter) [132]	Glial and neuronal expression (rat promoter in [132], human promoter in transgenic mice in [154])
gfaABC_1_D	Selected elements of the human *GFAP* gene promoter	High [155]	At least 9 weeks [155]	Expression pattern similar to that of full-size GFAP promoter in transgenic mice [154]; up to 98% of transgene-labeled cells were Aldh1L+ astrocytes in monkey (AAV5 delivery) [155]

^1^ NeuN—neuronal marker; GABA, GAD67—markers of all GABAergic neurons; PV, SST—markers of specific subtypes of GABAergic neurons; Aldh1L—astrocyte marker. A small fraction of cells immunopositive for these markers may belong to other cell types.

**Table 5 cells-14-00236-t005:** Properties of non-promoter regulatory elements described in Section 2.3.2 and Section 4.

Type of Element	Expression Efficiency	Cell Type Specificity (Examples)	Comments
Enhancers	Depends on promoter	For different enhancers, selective expression in GABAergic neurons in the forebrain was shown in [56,181]; in PV+ cortical interneurons in [17]; in particular subsets of neurons in the entorhinal cortex in [43]	Cell type-specific enhancers are usually much shorter than promoters with the same specificity
miRNA binding sites	Depends on promoter	For different elements, a shift towards expression in astrocytes was shown in [184]; selective expression in GABAergic neurons in [14,150]	Provide cell type-specific regulation on the posttranscriptional level
WPRE	Increases efficiency for some (intronless) promoters [42,187]	Non-specific	A shortened version of WPRE is also effective [188]
RNA polyadenylation switch	Translation is turned on when tetracycline is present	Non-specific	Provides regulation on the posttranscriptional level; an alternative to drug-regulated elements that does not introduce foreign proteins [160]

## Data Availability

No new data were created.

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
