# Peer review of "Regulatory Elements for Gene Therapy of Epilepsy"

_cells, 2025, doi:10.3390/cells14030236_

Round 1

Reviewer 1 Report

Comments and Suggestions for Authors

This review manuscript provides a comprehensive examination of regulatory elements in gene therapy for epilepsy, a field with significant therapeutic potential. The authors have done a commendable job of summarizing various promoters, enhancers, and other cis-regulatory elements. In general, the manuscript is well-crafted and easy to understand.

My comments are the following:

1. The abstract provides a good overview but lacks specific conclusions or key findings. It might be more beneficial to include a sentence summarizing the comparative advantages of different regulatory elements.

2. I strongly recommend including a visual summary (diagram) to illustrate key regulatory elements in epilepsy, which would enhance the review's clarity and comprehensiveness.

3. The detailed descriptions of promoters and enhancers are excellent. However, the comparisons between their effectiveness could be more structured. The authors may include a table summarizing the advantages and limitations of key promoters (e.g., CMV, SYN, NSE) in terms of tissue specificity, expression levels, etc.

4. Table 1 is a useful summary of promoters and transgenes but could be expanded to include enhancers and other regulatory elements.

Author Response

We thank the reviewer for the thorough reading of our manuscript and the suggested improvements.

Comments 1: The abstract provides a good overview but lacks specific conclusions or key findings. It might be more beneficial to include a sentence summarizing the comparative advantages of different regulatory elements.

Response 1: We rewrote the abstract to make it more relevant for the text of the paper, listing types of promoters and their properties that we compare in the text.

Comments 2: I strongly recommend including a visual summary (diagram) to illustrate key regulatory elements in epilepsy, which would enhance the review's clarity and comprehensiveness.

Response 2: We added a diagram (figure 1) depicting the common elements of expression vectors described in our review.

Comments 3: The detailed descriptions of promoters and enhancers are excellent. However, the comparisons between their effectiveness could be more structured. The authors may include a table summarizing the advantages and limitations of key promoters (e.g., CMV, SYN, NSE) in terms of tissue specificity, expression levels, etc.

Response 3: We added two separate tables summarizing data for universal and cell type-specific promoters (tables 3 and 4).

Comments 4: Table 1 is a useful summary of promoters and transgenes but could be expanded to include enhancers and other regulatory elements.

Response 4: We added a separate table (table 5) summarizing data for non-promoter elements.

Reviewer 2 Report

Comments and Suggestions for Authors

General Comments

I was pleased to read this very detailed review of gene therapies against epilepsy, for which there is a notable gap in the current literature. The use of the English language has been appropriate, and the references cited by the authors have been reasonable, with some being quite fresh.

My main concern lies in the high plagiarism score of the draft. Part of it stems from the names of vectors, promoters, enhancers, etc. (which cannot be addressed), but the authors should paraphrase other parts to minimize the overall score. There is also an issue with the article’s readability, where its full coverage of the topic (with more than 25,000 words in 44 journal-style pages) becomes its Achilles' heel. This can be circumvented by drawing more solid conclusions and providing summaries and tables at the end of sections; tasks that the current draft does not accomplish. Adding tables will enhance the accessibility of the information and allow readers to conveniently compare different approaches discussed within the text. As a result, I believe that the manuscript will be eligible for publication after undergoing a major revision to address plagiarism and add more tables (detailed below).

Specific Comments

Major Issues

1. As mentioned, plagiarism is a major issue with this draft (especially within section 1), and the authors should mitigate it by paraphrasing the corresponding sections.

2. p. 1, section 1.1: The definitions of epilepsy and seizure types are not accurate and should be re-stated according to the ILAE guidelines.

3. The authors should add several tables to summarize the salient points of cardinal sections. I would suggest, for example, adding a table for cell type-specific promoters (section 2.2, e.g., SYN, NSE, CaMKII, GAD67, GFAP, etc.), to demonstrate features of their design, types of cells in which they are active, in vitro and in vivo experiments they have been used in and the outcomes, and their limitations. This could also be applied to universal promoters (section 2.1, to depict their promising outcomes) and non-promoter cis-regulatory elements (section 4).

4. There is a limited number of small-scale clinical trials of gene therapies against epilepsy syndromes (e.g., NCT06063850, NCT06112275, and NCT04601974). In line with my previous comment, it would be appropriate to provide a table of the methodology and characteristics of the gene therapy approaches these trials have used.

Minor Issues

1. Per journal guidelines, abbreviations needed to be written in full at their first appearance within the text, with abbreviations in brackets; however, this has been violated several times. Here are some examples:

·        p. 2,  l. 79: AMPA is not defined

·        p. 4, l. 175: WPRE is not defined

·        p. 8, l. 287: HEK293 is correct

·        p. 9, l. 330: i.c.v. is not defined

·        p. 11, l. 456: NSC is not defined

·        p. 9, l. 348-350; p. 13, l. 516; p. 15, l. 630; p. 17, l. 744; p. 20, l. 905; p. 23, l. 1016; p. 27, l. 1233; p. 28, l. 1308; p. 40, l. 1870, etc.: Abbreviation must be within brackets.

2. The first sentence of the abstract is general and vague and should be replaced. The abstract also mentions poor communication between different specialists, which does not sound very relevant to the text. Overall, I would suggest re-writing the abstract with a focus on the pros and cons of currently available cis-regulatory elements and advances in the field.

Author Response

We thank the reviewer for the thorough reading of our manuscript and the suggested improvements. We performed a major revision as was advised, reworded section 1 and added a few more tables to better summarize the data.

Comments 1: As mentioned, plagiarism is a major issue with this draft (especially within section 1), and the authors should mitigate it by paraphrasing the corresponding sections.

Response 1: We reworded some sentences in this section as requested (and we also added a few more citations). A review paper is by definition a compilation of material from other papers, and for some statements (like the official ILAE definitions) paraphrasing would decrease accuracy. We also consider it necessary to start with definitions of the key concepts we describe and to cite, in addition to experimental papers, a few reviews with similar topics, especially in the introduction. This means the introduction text has a lot of similarity with other reviews about epilepsy and gene therapy; however, we cite the source of every statement that was not our own conclusion.

Comments 2:  p. 1, section 1.1: The definitions of epilepsy and seizure types are not accurate and should be re-stated according to the ILAE guidelines.

Response 2: We added citations of three ILAE papers about the definition and classification of seizures, the clinical definition of epilepsy and the current classification of epilepsy syndromes.

Comments 3:  The authors should add several tables to summarize the salient points of cardinal sections. I would suggest, for example, adding a table for cell type-specific promoters (section 2.2, e.g., SYN, NSE, CaMKII, GAD67, GFAP, etc.), to demonstrate features of their design, types of cells in which they are active, in vitro and in vivo experiments they have been used in and the outcomes, and their limitations. This could also be applied to universal promoters (section 2.1, to depict their promising outcomes) and non-promoter cis-regulatory elements (section 4).

Response 3: We added 3 separate tables summarizing data for universal promoters (table 3), cell type-specific promoters (table 4) and non-promoter elements (table 5).

Comments 4:  There is a limited number of small-scale clinical trials of gene therapies against epilepsy syndromes (e.g., NCT06063850, NCT06112275, and NCT04601974). In line with my previous comment, it would be appropriate to provide a table of the methodology and characteristics of the gene therapy approaches these trials have used.

Response 4: We added a table (table 1 in the revised manuscript) summarizing the data about 4 announced clinical trials in addition to citing them in the text. In the context of our review, the most important information about these trials is the elements used in the viral vector and their functions. The information aimed at medical specialists (participation criteria, cohorts, outcome measures etc.) would be excessive in our paper.

Comments 5:  Per journal guidelines, abbreviations needed to be written in full at their first appearance within the text, with abbreviations in brackets; however, this has been violated several times. Here are some examples:

  • p. 2, l. 79: AMPA is not defined
  • p. 4, l. 175: WPRE is not defined
  • p. 8, l. 287: HEK293 is correct
  • p. 9, l. 330: i.c.v. is not defined
  • p. 11, l. 456: NSC is not defined
  • p. 9, l. 348-350; p. 13, l. 516; p. 15, l. 630; p. 17, l. 744; p. 20, l. 905; p. 23, l. 1016; p. 27, l. 1233; p. 28, l. 1308; p. 40, l. 1870, etc.: Abbreviation must be within brackets.

Response 5: We fixed this issue in the specified places and in some other places (the line numbers are different in the revised version). In one case (small guide RNA) the full name was an unnecessary repeat since the abbreviation has been already deciphered earlier.

 Comments 6:   The first sentence of the abstract is general and vague and should be replaced. The abstract also mentions poor communication between different specialists, which does not sound very relevant to the text. Overall, I would suggest re-writing the abstract with a focus on the pros and cons of currently available cis-regulatory elements and advances in the field.

Response 6: We rewrote parts of the abstract as advised.

Round 2

Reviewer 2 Report

Comments and Suggestions for Authors

I have no further concerns.